

# Investigation of the effect of Reynolds number and inflow parameters on mean and turbulent flow over complex topography

Ryan Kilpatrick[1,3], Horia Hangan[1,2], Kamran Siddiqui[1,3], Dan Parvu[1,2], Julia Lange[4], Jakob Mann[4], and Jacob Berg[4]

[1]WindEEE Research Institute, University of Western Ontario, London, Ontario, N6M 0E2, Canada
[2]Department of Civil and Environmental Engineering, University of Western Ontario, London, Ontario, N6A 5B9, Canada
[3]Department of Mechanical and Materials Engineering, University of Western Ontario, London, Ontario, N6A 5B9, Canada
[4]DTU Wind Energy, Technical University of Denmark, Frederiksborgvej 399, DK-4000 Roskilde, Denmark

*Correspondence to*: R. Kilpatrick (rkilpatr@uwo.ca)

**Abstract.** A characterization of mean and turbulent flow behaviour over complex topography was conducted using a large-scale (1:25) model of Bolund Hill in the WindEEE Dome at Western University. The specific topographic feature considered was an escarpment. A total of eight unique inflow conditions were tested in order to isolate the impact of key parameters such as Reynolds number, inflow shear profile and upstream effective roughness, on flow behaviour over the escarpment.

The results show that the mean flow behaviour was generally not affected by the Reynolds number, however a slight increase in speed-up over the escarpment was observed for cases with lower upstream roughness. The shape of the inflow wind shear profile also had a minor impact on the mean flow near the escarpment. More significant effects were observed in the turbulent flow behaviour, where the turbulent kinetic energy (TKE) over the escarpment was found be a strong function of upstream roughness and a weak function of the Reynolds number. The local change in the upstream wind shear was found to have the most significant influence on the TKE magnitude, which more closely approximated the full-scale TKE data, and had not been previously observed in wind tunnel modelling of this topography.

## 1 Introduction

Wind turbines over the last few decades have emerged as a reliable and cost-competitive means of producing clean, renewable electricity. Although typically built on relatively flat terrain such as plains and farmland, wind farms are increasingly being placed in more rugged, or complex, terrain, marked by abrupt changes in elevation (Palma et al., 2008).

These sites often have strong wind resources, yet designing wind farms for these regions involves additional challenges due to the changes imposed by the terrain on the three-dimensional structure of the wind, such as speed-up regions, changes to the wind shear profile, large vertical wind velocities, and modification of turbulence characteristics (Walmsley and Taylor, 1996; Botta et al., 1998). As a result, the essential prediction of on-site wind conditions, often estimated from measurements at a limited number of mast locations, also becomes more challenging. The use of linearized models, the current industry standard for wind resource assessment and turbine micro-siting, proven to be very effective in gently sloping





terrain, can produce inaccurate results when applied at sites with very complex terrain (Palma et al., 2008; Berg et al., 2011). The use of more advanced modelling techniques such as Reynolds Averaged Navier Stokes (RANS), and Large Eddy Simulation (LES), have generally proven to be more accurate in complex topographic terrain (e.g. Rasouli and Hangan, 2013) compared to field measurements, and are making inroads with industry, although they come with the trade-off of

higher computational cost (Ayotte, 2008; Ayotte et al., 2010). These advanced models generally require a higher degree of user input and experience and thus results can be significantly affected by changes to boundary conditions, turbulence closure models and other parameters, as shown for example in the wide spread of CFD results in the Bolund blind comparison exercise (Bechmann et al., 2011).

Thus, a better understanding of the wind regime in complex terrain, from a fundamental fluid dynamics perspective, is

critical, given the opportunities for improved overall wind turbine performance including higher annual electricity production and reduced fatigue loading and associated maintenance costs (Peinke et al., 2004; Berg et al., 2011). This improved understanding of the flow behaviour can be used by modellers to select appropriate boundary conditions and turbulence models with greater confidence. One area that is not fully understood, and forms the subject of this study, is the sensitivity of the mean and turbulent response of the flow over complex topography to changes in the inflow conditions.

**1.1 Wind tunnel modelling of flow over topography**

In order to improve computational models, the model results need to be validated against actual flow conditions. Full-scale testing is ideal for this purpose, however due to the lack of control of inflow conditions, significant testing cost, time and effort required, wind tunnel modelling has served as a valuable tool for development and validation of both numerical and analytical models (Ayotte and Hughes, 2004). Provided that certain conditions are met, measurements taken of the flow

across a scale model can provide very useful and repeatable representations of full scale conditions as well as benchmarking for the validation of numerical and analytical models. The controlled environment of the wind tunnel provides a means of isolating the effects of various parameters on the mean and turbulent flow behaviour, which is usually not possible in the field.

There are several examples of wind-tunnel experiments conducted on flow over scale models of real topography for the

purpose of wind resource assessment and wind turbine siting. These include isolated hill cases such as Askervein Hill (Taylor and Teunissen, 1987), Kettles Hill (Salmon et al., 1988), and more recently Bolund Hill (Berg et al., 2011), as well as highly topographically complex regions with multiple hills and valleys (Chock and Cochran, 2005; Rasouli et al., 2009; McAuliffe and Larose, 2012).

**1.2 Bolund experiment**

The Bolund experiment arose from the need for additional model validation of flow in a complex terrain, extending the Askervein Hill Project of the early 1980s by offering steeper terrain and thus a greater challenge for numerical models to resolve. Bolund Hill is a peninsula located near Roskilde, Denmark, and is characterized by a long upstream open fjord



fetch, a steep escarpment and a long flat section on top of the island. The Bolund topography is geometrically similar to a typical wind turbine site in complex terrain, albeit at smaller scale, and is well-suited as a test site given its well-defined, undisturbed inflow conditions, neutral atmospheric stratification and relative absence of thermal and Coriolis effects (Berg et al., 2011). Although Bolund is a small hill, approximately 12 m high by 75 m wide and 130 m long, similarity laws allow for

upscaling of 10 – 30 times under neutral atmospheric stratification.

Studies of the wind flow over Bolund Hill include the original field campaign (Berg et al., 2011); follow-up Lidar measurements of the escarpment wake (Lange et al., 2016); computational and physical modelling of the hill as a part of the blind comparison test (Bechmann et al., 2011); wind tunnel modelling (Yeow et al., 2015); LES modelling (Diebold et al., 2013); and wind-tunnel and LES modelling (Conan et al., 2016). During the field campaign, measurements were taken via

35 anemometers on 10 masts, positioned along two main incoming flow directions referred to as, Line A (239°) and Line B (270°). These were the benchmark measurements against which the results of subsequent modelling efforts have been compared. A detailed diagram of the Bolund topography, with mast positions and flow directions, appears in Berg et al. (2011).

### 1.3  Present study: characterization of mean and turbulent flow over Bolund across a range of input conditions

The present study is focused on the characterization of the flow over Bolund Hill, along Line B, in the vicinity of the escarpment, using two physical scale models (1:100 and 1:25), at Reynolds numbers (based on model hill height and wind speed at hill height) ranging from $4\times10^4$ to $5\times10^5$. The main objectives of this study were to isolate and analyse the dependence of the mean and turbulent flow behaviour over the escarpment on various parameters including Reynolds number, inflow wind shear profile, and upstream roughness. The 1:25 scale experiments were conducted at the Wind

Engineering, Energy, and Environment Research Institute (WindEEE), while the 1:100 scale experiments were conducted at the Boundary Layer Wind Tunnel Laboratory (BLWTL). Both facilities are located at Western University. The two sets of results were then compared with the full-scale measurements, and with results from previous studies on Bolund Hill. Particle Image Velocimetry (PIV) and Cobra Probes were used for flow velocity measurements at key locations on the scaled models.

## 2 Experimental setup

### 2.1 WindEEE experimental setup

### 2.1.1 WindEEE facility

The WindEEE dome is a unique wind research facility designed to simulate a wide variety of wind flow patterns including rotational (axisymmetric) and boundary-layer flows at larger laboratory scales. A general description of the facility is

provided in Hangan (2014). The hexagonal test chamber (25 m diagonal length), is enclosed in a return air chamber of 40 m





diagonal length. The WindEEE test chamber contains 106 fans, whose wind speed and direction can be varied independently to produce the desired flow conditions. The facility can be operated in two distinct modes: multi-fan wind tunnel or axisymmetric mode. The present experiment was conducted exclusively under the former configuration, with only the fans along one wall of the hexagon mounted in an array format i.e. four rows of 15 fans each, for a total of 60 fans. Each of these

fans are 0.8 m in diameter and operate at approximately 25 m/s at nominal power of 30 kW. Each fan is equipped with variable speed drive and can be individually controlled to create a customized flow pattern. A contraction section was positioned immediately downstream of the 60-fan wall to improve flow uniformity and increase flow speed across the 5 metre diameter test section (turntable) in the centre of the chamber. A trip and a series of spires were employed upstream to enhance turbulence intensity.

In addition to individual fan control, the WindEEE facility also provides the ability to set roughness element position and height, allowing physical simulation of a wide range of incoming Atmospheric Boundary Layer (ABL) flow profiles. There are over 1500 roughness elements in the test chamber, each with maximum height of 30 cm. The present experiment employed only the roughness element sections in the vicinity of the contraction, upstream of the turntable. Two different roughness element configurations, both with uniform element height of roughly 7.5 cm, were used for the WindEEE

experiment, hereinafter referred to as RC1 and RC2. For RC1, all of the roughness elements upstream of the turntable were raised, whereas for RC2, one block of about 80 elements immediately upstream of the turntable was lowered to the floor, resulting in a lower effective roughness value than that for RC1.

### 2.1.2 Bolund scale model

The 1:25 scale model of Bolund Hill was produced by CNC milling of several large blocks of Expanded Polystyrene (EPS)

according to topographical data of the island. These blocks were then glued together and painted black with latex paint. The overall size of the model was roughly 4.5 m across, 0.5 m high, and 3.5 m long. A solid ramp with slope of roughly 45° was constructed from EPS and fastened to the downstream edge of the model to provide a smooth transition and reduce unwanted flow separation. The model was positioned in the chamber such that the escarpment edge was roughly 12.4 m from the 60-fan wall, and the plane of measurement (Line B) was parallel to the flow direction. A photograph of the experimental setup

is presented in Figure 1a. The $X = 0$ position in the longitudinal direction is indicated by the centre-point of the model, $C_p$, also the intersection of Line A and Line B, as per the full-scale co-ordinate system, shown in Figure 1a.

### 2.1.3 PIV measurement

Particle Image Velocimetry (PIV) was used to measure the two-dimensional velocity field in a vertical plane above the model, along line B, in the vicinity of the escarpment. The measurement region encompassed a rectangular area extending

roughly from $Z = 11.4$ m to 25 m and $X = $ -70 m to $X = $ -20 m in the full-scale Bolund co-ordinates, where the value $Z = 0$ corresponds to sea level. Throughout this work uppercase $Z$ denotes absolute height above sea level while lowercase $z$ denotes height relative to the hill surface. Three 12 megapixel cameras (IO Industries Flare 12M125-CL), each with 105 mm



f/2D Nikon AF DC-NIKKOR lenses, were used to capture images. The cameras were positioned in a row parallel to the flow direction, facing the model at a distance of roughly 3.55 metres from the camera lens to the measurement plane, at a height such that the bottom of the camera frame of view was just below the hill surface. Camera resolution was 4096 x 3072 pixels, and the corresponding measurement field of view for each camera in the current setup was about 0.78 m wide by 0.58 m

high. The horizontal positions of the cameras were set in a way to form overlap among the adjacent fields of view to ensure spatial continuity of flow measurements. The overlap between cameras 1 and 2 was 0.167 m, and between cameras 2 and 3 was 0.088 m, with camera 1 being the most upstream. Thus the combined measurement area was roughly 2.09 m wide by 0.58 m high.

A Litron Nano Piv Series dual cavity Nd:YAG laser with the energy of 425 mJ/pulse operating at the wavelength of 532

nm was used to illuminate the flow field. The laser was positioned directly behind the model, pointing upstream, coincident with Line B, with the laser head roughly 0.60 m off the ground, as shown in Figure 1. A 50° cylindrical lens was positioned immediately in front of the laser head to convert the beam into a two-dimensional sheet. The laser was synchronized to the cameras and the frame grabber. In this study, the pulse repetition rate for each laser cavity was set at 9 Hz resulting in the image acquisition rate of 18 frames per second by each camera or 9 Hz for each image pair. The images were acquired via

Coreview software (IO Industries) as 8-bit grayscale images in the TIFF format. PIV data were recorded for 5 minutes per test case, providing roughly 2700 image pairs. An Ultratec CLF-4460 commercial fog generator, positioned in the dome's upper plenum, was used to seed the test chamber with non-toxic, water-based smoke that served as the tracer.

### 2.1.4 Cobra Probe measurement

Cobra Probes, manufactured by Turbulent Flow Instrumentation Pty Ltd., are dynamic multi-hole pressure probes for

measuring all three components of mean and fluctuating velocities and static pressure. In the present experiment, Cobra Probe measurements were taken at an upstream reference location, as well as a few positions along the hill. A vertical array of eight Cobra Probes was used, with spacing between probes ranging from roughly 5 cm near the bottom of the array to 15 cm near the top. The total vertical measurement distance was about 60 cm, or 15 m in full-scale. The upstream reference position was located 4.44 m upstream of $C_p$ in the model scale, or $X = -111$ m in the full-scale. Although initially intended

for the Cobra probe position to coincide with the full-scale upstream reference mast M0, located at $X = -180.8$ m, this was not possible in the current setup due to the proximity of the model to the contraction opening. However, the current location was deemed to be sufficient as a reference location given that it was far enough from the fan wall to assume fully mixed flow and far enough from the model to avoid significant slow-down effects. Along the hill, Cobra Probe measurements were taken at the escarpment edge ($X = -54.7$ m), at M6 ($X = -46.1$ m) and at M3 ($X = 3.2$ m), where values in parenthesis are full-scale

co-ordinates. Due to time constraints, Cobra Probe measurements at these positions were not taken for each of the PIV test case configurations. The probe array was mounted either on a stationary floor rack, or fixed to the overhead rail system, and moved to various positions along line B. All Cobra Probe measurements were conducted at acquisition rate of 10,000 Hz, output to file rate of 1250 Hz, and sampling time of 120 seconds.





## 2.2 BLWTL experimental setup

The model used for the BLWTL experiments was a 1:100 scale model of Bolund Hill, using the same topographical data as the WindEEE model. The BLWTL model was similarly cut from EPS, in two sections, and fastened together. The model was then fixed to the turntable at the centre of the test section and rotated such that the principal flow direction coincided with Line B (270° wind direction). Figure 1b shows a photograph of the experimental setup with wind direction and mast

positions indicated. BLWTL Tunnel 1 is an open circuit type with a length of 33 m and has the cross-section of 2.4 m (width) × 2.15 m (height) at the test section. In the present setup, three triangular spires, as well as a bar trip were positioned at the far upstream end of the tunnel, however no active roughness elements were used, in order to simulate upstream conditions with ABL profile over a smooth surface. The measurements were conducted at a wind speed of 4.6 m/s that

corresponds to a Reynolds number of approximately $3.6 \times 10^4$, based on the maximum height of the hill (0.117 m in model-scale) as the characteristic length scale.

Cobra Probe measurements were conducted along vertical profiles at model-scale positions equivalent to the full-scale co-ordinates of M0, M7, M6, M3 and M8, as well as at the escarpment ($X$ = -54.7 m in full-scale). The vertical velocity profile was obtained by using two Cobra Probes mounted to the wind tunnel traverse system and the vertical position was

incremented after each sample, by 12.7 mm in model-scale, or 1.27 m in full-scale, near the floor, and 50–100 mm (5–10 m full-scale) higher up. The vertical extent of the measurements was about 1.2 m from the floor (equivalent to 120 m in full-scale). PIV measurements were conducted as well, but are not presented here due to a number of issues with data quality.

## 3 Data processing

### 3.1 PIV data processing

PIV instantaneous velocity fields were obtained by cross-correlating the interrogation regions in the first image of the image pair with the corresponding search regions in the second image. An in-house algorithm implemented in image processing software Heurisko® developed by AEON Verlag & Studio GmbH & Co. KG was used for the PIV data processing. The search window and interrogation window sizes were set as 128 and 64 pixels, respectively, while grid size was 16 pixels. Spurious vectors were identified and corrected using a local median test developed by Siddiqui et al. (2001). Subsequently,

mean fields were calculated by averaging the respective velocity component (streamwise, $u$, and vertical, $w$, in the present case) at each grid point over the sampling time. The turbulent velocity fields were computed by subtracting the mean velocity from the instantaneous velocity at each grid point in a given velocity field. These two steps were performed using an in-house code in MATLAB.

PIV statistics were calculated in a manner analogous to Yeow et al. (2015) to enable direct comparison. Mean flow

speed, S, was calculated using the two mean wind components U and W from the PIV measurement plane (see Figure 1) as per:





$$S = (U^2 + W^2)^{1/2} \tag{1}$$

Results shown throughout this work are often expressed as a normalized speed-up ratio $S(x,z)/S_0(z)$, where $S_0(z)$ is the upstream reference speed at the same height. Since the upstream reference measurements were taken with Cobra Probes at a limited number of heights, extrapolation to all of PIV grid heights was performed using the logarithmic law (Manwell et al., 2009):

$$U(z)/U(z_r) = \ln\left(\frac{z}{z_0}\right) / \ln\left(\frac{z_r}{z_0}\right) \tag{2}$$

where $z_r$ is the reference height. Mean turbulent kinetic energy (TKE), $\overline{k}$ was calculated according to:

$$\overline{k} = \left(\overline{u'^2} + \overline{w'^2}\right)/2 \tag{3}$$

where lowercase $u'$ and $w'$ represent the fluctuating velocity vectors. The change in TKE, or TKE increment $\Delta\overline{k}$ was obtained by subtracting the upstream reference TKE $\overline{k}_{05}$ at a fixed height of $Z = 5$ m in full-scale (0.2 m in model-scale) from the measured TKE at each PIV grid position and normalizing by the square of the upstream reference speed, again consistent with Yeow et al. (2015) as per:

$$\Delta\overline{k} = \left[k(x,z) - \overline{k}_{05}\right]/S_0{}^2(z) \tag{4}$$

Despite efforts to properly align the three cameras, some minor discrepancies were observed in the velocity data recorded by each camera. For mean wind speeds, error between camera frames typically ranged from about 2–4%, with slightly more error in the highly turbulent region close to the escarpment and just above the model surface. To improve the clarity of presentation, a frame stitching algorithm using was implemented to smooth the data within the overlap region between camera frames. At each point in the overlapping region, a weighted average of the data at the two overlapping nodes was taken, such that data points closer to one camera or another were weighted more heavily towards that camera's values. The weightings varied linearly from 0.5 for each camera at the centre of the overlap region (equal weighting), to 1 and 0 on one side, and 0 and 1 on the other.

### 3.2 Cobra Probe data processing

Cobra Probe output data is generated by the companion TFI Device Control software, and consists of a time history of $u$, $v$ and $w$ wind speed components, as well as a summary output of the mean wind speeds, Reynolds stresses and pressures. Generally only the mean data from the summary files were used, with spot checks performed against the time history data to ensure consistency. The Cobra Probe results presented in this work generally use two-component calculations, where the spanwise wind speed component $v$ is neglected, as per Eq. 1 and Eq. 3, which allows for direct comparison with the PIV results, which is analogous to the approach adopted by Yeow et al. (2015) for hot-wire measurements. However, when comparing the WindEEE and BLWTL inflow profiles, measured with Cobra Probes, against the full-scale data from upstream mast M0, all three wind velocity components from the Cobra Probe data were used.



## 4 Inflow profiles

### 4.1 WindEEE inflow profiles

#### 4.1.1 Fan configuration

For the present set of experiments, the 60 fans were operated using four different configurations, which were selected in
attempt to match the full-scale incoming wind profile, as well as to produce a range of Reynolds numbers (see Table 1):

   a)   All fans running at 20% of the maximum fan RPM

   b)   All fans running at 30% of the maximum fan RPM

   c)   All fans running at 50% of the maximum fan RPM

   d)   Fans in row 1, 2, and 4 running at 50%, fans in row 3 at 75%, of the maximum fan RPM. For reference, fan row 1 is
        at floor level

The notation for each test case was set based on the upstream wind speed and the upstream surface roughness. That is, each
of the four fan configurations are identified by the mean streamwise incoming wind speed at the model escarpment height in
m/s (i.e. U5, U8, U14, U15) and the two upstream roughness configurations as RC1 and RC2, where, RC1 corresponds to
the higher upstream roughness. For example, case U5RC1 correspond to the test case conducted at the inflow condition of 5
m/s wind flowing over higher roughness. These combinations yielded eight unique flow configurations representing the
WindEEE PIV test cases described throughout this work, as listed in Table 1.

#### 4.1.2 Calculation of Reynolds number, friction velocity and surface roughness

The inflow parameters for the WindEEE experiment in Table 1 were determined from the upstream reference Cobra Probe
data. Reynolds number was calculated according to:

$$Re = \frac{U_h h}{\nu} \tag{5}$$

where the characteristic length $h$ is the hill height, $U_h$ is the inflow streamwise velocity at $h$, and $\nu$ is the kinematic viscosity
of air at 20°C i.e. $1.50 \times 10^{-5}$ m/s$^2$. The Reynolds numbers in Table 1 use the model hill height $h = 0.468$ m. For the test cases
with the same fan configuration but different roughness configuration, Reynolds number was almost identical.

For the present study, which focuses mainly on how upstream conditions affect flow behaviour over the escarpment, the
means by which the upstream parameters are calculated are important, as there is often some variability depending on the
method of calculation. For example, friction velocity $u_*$ can be determined using several different methods, which often
show considerable differences between them (Weber 1999). To compare the variability of the resulting normalized upstream
profiles for the WindEEE experiment, the friction velocity was calculated using four different methods. For method 1,
friction velocity was calculated according to Eq. 6, as per Weber (1999) using only the longitudinal component of the
Reynolds stress vector, which is the same method used by Berg et al. (2011) to calculate friction velocity using data from the
upstream reference mast M0 in the Bolund field campaign.





$$u_\bullet = (-\overline{uw})^{1/2} \tag{6}$$

Method 2 adds the spanwise Reynolds stress component (Ly 1993, Weber 1999), and always produces a higher value of $u_\bullet$ than Method 1. It is similar to the method used in Bechmann et al. (2011) and is given by:

$$u_\bullet = [(\overline{u'w'})^2 + (\overline{v'w'})^2]^{1/4} \tag{7}$$

For Methods 1 and 2, a single reference value $u_{\bullet 05}$ was taken as the friction velocity at a reference height of $Z = 5$ m in full-scale (0.2 m in model-scale), consistent with the approach used by Bechmann et al. (2011) and Yeow et al. (2015). For method 3, upstream effective roughness $z_0$ and friction velocity $u_\bullet$ were determined by fitting the streamwise velocity profile data within the logarithmic region to the standard logarithmic wind profile for neutral stability conditions (Manwell et al., 2009):

$$U(z) = \frac{u_*}{\kappa} \ln\left(\frac{z}{z_0}\right) \tag{8}$$

where $z$ is the vertical height from the ground, $U(z)$ is the streamwise wind speed at that height, and the von Karman constant $\kappa$ was considered to be 0.41. Method 4 may be considered a combination of Methods 1 and 3, and follows the approach of Akomah et al. (2011), who describe a region of constant shear stress corresponding to the equilibrium sub-layer where TKE production and dissipation balance. The values of friction velocity computed using each of the four methods are presented in

Table 1. The difference between the highest and lowest estimate was relatively high, ranging from about 15% to 50% depending on the test case. Friction velocity is still calculated using Eq. 6, however unlike in Method 1 where a single data point was used, Method 3 uses the mean of the values within the constant shear stress region, which were identified from the plots of height vs. $|\overline{u'w'}|$ as the first three data points closest to the floor for each test run. The $z_0$ values were obtained using Method 3, and are presented in full-scale units in Table 1. The values show a clear distinction between the RC1 cases ($z_0 \sim$

$10^{-3}$ m) and RC2 cases ($z_0 \sim 10^{-6}$ m). The full-scale roughness measured at mast M0 was $z_0 = 3\times10^{-4}$ (Berg et al., 2011), so the U15RC1 case shows the closest match.

### 4.1.3 Comparison between inflow profiles

Cobra Probe measurements of upstream reference mean flow speed $S_0$ and TKE $\overline{k_0}$, normalized by the $u_\bullet$ values estimated using method 1, are shown in Figure 2a, along with the full-scale measurements at M0 (Berg et al., 2011). A clear separation

is observed between the profiles with higher roughness (RC1) and those with lower roughness (RC2), with the RC2 group having higher normalized mean wind speed as well as TKE. Comparison of the upstream mean speeds for the test cases with the full-scale data shows that all model-scale values are lower than the full-scale wind speeds, with the exception of U5RC2, whereas the normalized TKE profiles are all higher than the full-scale profiles, illustrating the inherent difficulty in matching both the mean wind speed and TKE profiles with the full-scale values. The shape of the TKE profiles is in contrast to the

wind-tunnel experiment conducted by Yeow et al. (2015) whose normalized TKE inflow profiles were lower than the full-scale values, and decreased with height. Most of the WindEEE normalized TKE profiles are relatively vertical between $z = 5$



m and $z$ = 12 m which is consistent with the full-scale data, although having only two full-scale data points available from the reference mast M0 in the field campaign, none of which were above a height of 12 m (i.e. just above escarpment height), is a limiting factor in determining whether a good match to the full-scale conditions has been achieved.

The results for Method 2 (not shown) are similar to Method 1, with all profiles shifted slightly to the left, given the slightly higher values of $u_*$. There is also less separation between the RC1 and RC2 groups. Inflow profiles determined using Method 3 are somewhat different, with the RC1 profiles showing a better match to the full-scale data for mean speed, although still higher for TKE, but significantly higher values for the RC2 group, for both mean speed and TKE, due to the higher values of $u_*$. Profiles using Method 4 are quite similar to those of Method 1, with profiles collapsing slightly more within the RC1 and RC2 groups.

## 4.2 BLWTL inflow profiles

Table 2 shows the main test parameters for the BLWTL Cobra Probe measurements. Friction velocity was calculated as per the four methods outlined above, and $z_0$ was estimated using Method 3. Figure 2b shows inflow profiles from the BLWTL Cobra Probe data, measured at the upstream reference location of $X$ = -1.82 m in model scale ($X$ = -182 m in full-scale). Mean speed and TKE were normalized by friction velocity calculated using the four methods identified above. The results show that the profiles for Methods 1–3 are quite close to each other, and higher than the full-scale data points, while Method 4, with higher $u_*$, produced profiles shifted slightly to the left, and matched particularly well with the full-scale data. The reduction in normalized TKE with height was consistent with the inflow profiles measured by Yeow et al. (2015), but different from the WindEEE and full scale TKE profiles, which were relatively constant with height over the measurement region.

## 5 Results and Discussion

The results are divided into two main sections: analysis of the mean flow behaviour, and analysis of the turbulent flow behaviour. Most of the results presented were obtained from the WindEEE PIV data, while some additional data is presented from the WindEEE Cobra Probe profiles, as well as the BLWTL Cobra Probe measurements as necessary.

### 5.1 Mean flow behaviour

The streamlines of the mean flow field are shown in Figure 3 for upstream velocities of U5, U8, U14 and U15 at higher roughness configuration (RC1). Mean streamlines for the RC2 cases (not shown) were nearly identical. None of the streamlines for the mean flow showed any clear evidence of recirculation. Even upon close inspection of the mean flow field in the vicinity of the escarpment, it is difficult to ascertain whether the negative values of streamwise velocity observed in the separated flow region are indicative of recirculation or due to other factors such as near-wall effects. The small region between -55.25 m < $X$ < -55 m and 11.75 m < $Z$ < 12 m appears to show some vectors of low magnitude pointing





horizontally in the reverse streamwise direction, however the higher uncertainty in this region, due to very high velocity gradients, prohibits drawing a firm conclusion.

Contour plots of speed-up ratio for U14RC1 (Figure 4a) and U15RC1 (Figure 4b) clearly illustrate the speed-up region near the escarpment, and the re-establishment of the boundary layer on top of the hill. The U5 and U8 contour plots (not
shown) were very similar to the U14 case, analogous to the similarity observed in the streamline plots between the three cases. While speed-up is generally similar between the U14RC1 and U15RC1 cases, slightly higher values are observed for U15RC1 in the vicinity of the escarpment, and this case also shows a more elongated, oblong shape of the speed-up region at the escarpment edge. Reynolds number for the two flows did not differ by a great amount ($4.57 \times 10^5$ for U14RC1 vs. $5.21 \times 10^5$ for U15RC1), i.e. much less than the difference in Reynolds number between the U5RC1 and U14RC1 cases,
indicating that the difference in normalized mean flow behaviour can be most likely attributed to the higher upstream shear for the U15 case.

### 5.2 Mean flow comparison to previous experiments

In addition to the full-scale measurements, results from previous physical modelling of the Bolund Hill, are available in the literature for comparison to the present results. These include wind tunnel and water channel experiments from the blind
comparison (Bechmann et al., 2011), wind tunnel PIV and 3-component hotwire (3CHW) tests conducted by Yeow et al. (2015) at 1:115 scale at two Reynolds numbers ($4.15 \times 10^4$ and $8.21 \times 10^4$), and wind-tunnel PIV modelling conducted by Conan et al. (2016) at 1:500 scale and $Re = 2.1 \times 10^4$. Benchmarking the WindEEE Cobra Probe and PIV, and BLWTL Cobra Probe results from the present experiment against these datasets provides some validation of the present experimental procedure, and also serves as an initial point of discussion on the differences between conducting the same experiment at
three scales, i.e. wind tunnel ($Re \sim 10^4$), WindEEE ($Re \sim 10^5$) and full-scale ($4.25 \times 10^6 < Re < 1.02 \times 10^7$).

Figure 5 shows horizontal profiles of the wind speed-up at two locations corresponding to the full-scale mast measurement positions at heights of $z = 2$ m (Figure 5a) and $z = 5$ m (Figure 5b) above hill surface level. Results from the WindEEE PIV data and those of previous experiments mentioned above are presented for comparison. The topography and the mast locations are shown in Figure 5c for reference. Figure 6 shows the comparison for vertical profiles at three
horizontal locations along the hill. The U14RC1 and U15RC1 cases were selected from the eight WindEEE PIV cases as representative cases to avoid clutter; the differences between all of the WindEEE cases are discussed further below. From the horizontal profiles, agreement is generally quite good between all datasets at $z = 5$m, whereas significant variability is observed at $z = 2$ m, which is within the highly turbulent shear layer observed in the TKE contour plots (see Figure 10), referred to also by Yeow et al. (2015) and observed in the scanning Lidar data (Lange et al., 2016). Similarly for the vertical
profiles, better agreement is observed at position M7 upstream of the escarpment (Figure 6a), with greater variability seen at the other two positions (Figure 6b, c), with higher variability at $z < 5$ m.



### 5.3 Influence of Reynolds number, upstream roughness, inflow profile and model resolution on mean flow

The WindEEE experiments were conducted by changing one variable at a time, allowing for the influence of a particular modifier to the flow to be isolated and the resultant flow behaviour analysed. In this section, the isolated effects of Reynolds number, upstream roughness, shape of the inflow profile, and scale model & measurement resolution, on the mean flow behaviour are discussed.

### 5.3.1 Reynolds number and upstream wind profile effects on the mean flow

The horizontal profiles of the wind speed-up for four Reynolds numbers at full-scale heights of $z = 5$ m and $z = 2$ m above the island surface level are shown at two upstream roughness cases; higher roughness RC1 in Figure 7a and lower roughness RC2, in Figure 7b. Full-scale data are also plotted for reference. The results show almost identical trends of the normalized mean flow for three uniform fan speed cases (U5, U8 and U14), for both RC1 and RC2. This indicates an absence of Reynolds number effect on the mean flow over a Reynolds number range of $1.7 \times 10^5$ to $4.6 \times 10^5$. The U15 case, however, with modified inflow shear profile, displays different behaviour than the uniform fan speed cases. The U15RC1 peak speed-up is higher at the escarpment compared to the other RC1 cases, then changes to become relatively lower further downstream. For the U15RC2 case, speed-up is generally equal or slightly lower than the other RC2 cases at $z = 5$ m (Figure 7c), and lower along horizontal locations at $z = 2$ m (Figure 7d).

In Figure 8, a similar comparison is made along the vertical profiles at M6. Results show a trend similar to that observed for the horizontal profiles, i.e. little difference among the mean flow profiles at three uniform fan speed cases, with the RC2 profiles collapsing more closely. Again the U15RC1 case (Figure 8a) shows different behaviour, with higher speed-up than the other cases, and also shows a better match to the full-scale data points. For RC2 (Figure 8b), the U15 case generally collapses with the others, with the only difference being the relatively lower speed-up for $z < 4$ m which is again closer to the full scale behaviour.

### 5.3.2 Upstream roughness effects on the mean flow

The comparison of speed-up profiles at the same Reynolds number but different roughness configuration provides an insight into the effect of upstream surface roughness, $z_0$, on the mean flow behaviour over the escarpment. Such analysis can be obtained by comparing the profiles in Figure 7 and Figure 8 for two roughness cases. It is observed at that in general, the speed-up profiles for the same Reynolds number at the two different roughness heights were similar, despite relatively significant difference between $z_0$ values, which was about three orders of magnitude larger for RCI compared to RC2. The difference in peak speed-up for two roughness configurations at $z = 2$ m was about 6% 8% and 5% for U5, U8 and U14, respectively. For the uniform fan speed cases, a lower upstream $z_0$ was found to generate a higher peak speed-up at the escarpment, with diminishing effect moving downstream. A different trend was observed for the U15 cases, where a slight reduction in peak speed-up of about 3% was observed at the escarpment for the RC2 case, with the difference between the





two roughness cases slightly growing moving downstream. At M6, the lower roughness cases showed a slightly better match to the full-scale data (see Figure 8).

### 5.3.3 Effect of measurement and model resolution on the mean flow

The Cobra Probe measurements were taken under identical fan speed and roughness element configurations as the PIV cases, although not simultaneously, and therefore from the mean flow perspective, they provide useful independent evidence for Reynolds number dependence. A comparison of speed-up ratio between PIV and Cobra Probe measurements, for the three uniform fan speed cases, for both roughness configurations, along the same vertical profile at M6, is shown in Figure 9. The results show very good matching between the two methods of measurements, with some systematic bias error resulting in slightly lower speed-up for Cobra Probe measurements, perhaps due to PIV calibration. Notwithstanding, very little evidence of Reynolds number dependence is observed between the Cobra Probe profiles, confirming the trends observed earlier in the PIV data.

Now turning to the discussion on the effect of model resolution on the mean flow, it is generally accepted by wind tunnel modellers that for bluff bodies submerged in deep boundary layers, Reynolds number effects are negligible for $Re > (2–3)\times10^4$, particularly for flows without steady vortical regions (Lim et al., 2007). Given that the BLWTL tests were conducted at Reynolds number above this threshold ($3.6\times10^4$), as were the two tests conducted by Yeow et al. (2015), at $4.15\times10^4$ and $8.21\times10^4$, one would expect to see Reynolds number independence preserved between normalized speed-up profiles at the BLWTL scale (1:100) and the WindEEE scale (1:25), measured using the same instrument, with similar upstream conditions. Such comparison can be made using results in Figure 6, which illustrates the speed-up profiles from Cobra Probe measurements at BLWTL and WindEEE. Some discrepancies are observed, particularly at M6 at $z < 5$ m, where WindEEE measurements were found to be higher, and closer to the full-scale measurements. Assuming that the Reynolds number threshold (Lim et al., 2007) is applicable under present conditions, it can be concluded that the discrepancies between Cobra Probe results observed at two different model resolutions, are not due to the Reynolds number but rather are caused by other factors related to model and measurement resolution such as proximity of the measurement instrument to the surface, size of the instrument relative to the model and surface roughness of the model.

### 5.4 Turbulent flow behaviour

The results for the turbulent flow are presented in a similar manner as for the mean flow behaviour in Section 5.1. To obtain a better insight into the overall turbulent flow behaviour, contour plots of the change in TKE $\Delta \bar{k}$ over the same area as in the earlier speed-up plots, are shown in Figure 10a and Figure 10b for U14RC1 and U15RC1, respectively. A high-turbulent intensity region is observed at the escarpment, which dissipates moving downstream. Several significant differences are observed between the two cases, with U15RC1 having a larger high-intensity TKE region near the escarpment, and a longer





and higher wake region. The TKE increment also begins further upstream of the escarpment. The U5 and U8 TKE contour plots (not shown) were similar to the U14 case, but with slightly lower values of $\Delta\bar{k}$ throughout.

**5.5 Turbulent flow comparison with previous experiments**

A comparison of horizontal profiles of WindEEE TKE increment against previous experimental results, at $z = 5$ m and $z = 2$ m above surface level is presented in Figure 11. The two WindEEE PIV profiles stand out from the others as they feature a shallow hump between M6 and M3 at $z = 5$ m, and a sharp spike between the escarpment and M6 at $z = 2$ m. Both features are much more pronounced for the U15RC1 case compared to U14RC1. The Cobra Probes were not able to capture the TKE spike to the same extent, also observed in the vertical profile at M6 (see Figure 12). As was the case for speed-up ratio, the U15RC1 case was observed to better approximate the full-scale values of $\Delta\bar{k}$ than the others.

**5.6 Influence of Reynolds number, upstream roughness, inflow profile and model resolution on turbulent flow**

**5.6.1 Reynolds number and upstream wind profile effects on the turbulent flow**

Figure 13(a,b) and Figure 13(c,d) show horizontal profiles of $\Delta\bar{k}$ for the four wind speed cases at RC1 and RC2, respectively. The two U15 profiles stand out from the other cases, and the discrepancy is much more significant than it was for the speed-up ratio – the peak TKE increment for U15RC1 was about 200% higher than that for U14RC1. As noted earlier, the only difference between the U14 and U15 cases was the increase in the operating speed of fans in the third row by 50% compared to all other fans. This produced a slightly higher Reynolds number at the hill height for the U15 cases than for U14, but the Reynolds number difference is only about one third of the difference between the U8 and U14 cases. Thus the difference in profile shapes at heights of $z = 2$ m and $z = 5$ m above the island appears to be attributable mainly to the induction of the strong shearing effect between fan rows 2 and 3, despite the fan row interface being about two metres off the tunnel floor in model scale, or 50 m in full-scale, more than four times the hill height. There do not appear to be any indications of this behaviour from the inflow profiles, at least not from the normalized mean wind speed and TKE profiles, where the U15RC1 inflow profile is generally similar to the others in the RC1 group, regardless of which method was used to determine $u_*$. While analyzing the WindEEE inflow profiles up to a height of about 100 m in full-scale would have provided a better picture of the difference in inflow conditions between the U14 and U15 cases, the comparison with upstream data from the field campaign at mast M0 at these heights not being available would preclude the interpretation of this better fit between the U15 case and full scale. Nevertheless, the results highlight the important fact that a relatively small change to the inflow wind shear profile, even well above the model height, can significantly alter the turbulent flow behaviour near the hill surface.

Among the three uniform fan speed profiles, there is little difference at $z = 5$ m, however, at $z = 2$ m, peak TKE for the U14 case between the escarpment and M6 is higher than the other two cases, for both RC1 and RC2, indicating a possible Reynolds number dependence in this region. Vertical profiles of change in TKE at M6 were also plotted (see Figure 14) for





the RC1 and RC2 cases. The profiles for the uniform fan speed cases again tended to collapse, with the exception of the U14 cases, below $z = 2$ m, where a slight increase in TKE was observed. Consistently higher TKE is observed for the U15RC1 case, and again the U15 results are closer to the full scale data.

### 5.6.2 Upstream roughness effects on the turbulent flow

The influence of upstream roughness can be seen in Figure 14 at M6. For the uniform fan speed cases, configuration RC2, with lower $z_0$, appeared to cause a moderate increase in TKE compared to RC1 at $z < 5$ m, while little difference in TKE was observed between the U15 cases. A comparison of horizontal profiles of RC1 and RC2 cases (not shown) showed higher TKE increment for the RC2 cases at all positions downstream of about $X = -50$ m, particularly at $z = 2$ m. The difference in peak TKE increment at different roughness configurations was about 13%, 28%, 27% at U5, U8 and U14, respectively, with
the RC2 value being higher for these cases. The U15 case showed negligible difference for the two roughness cases with a weak trend of a decrease in TKE increment with the roughness. Thus, a change in the upstream roughness was observed to have an effect on the TKE for the uniform fan speed cases, while the case with modified shear profile appeared to be more resilient to changes in the roughness.

### 5.6.3 Effect of model and measurement resolution on the turbulent flow

A comparison of TKE increment between WindEEE PIV and WindEEE Cobra Probe measurements, for the same Reynolds number, and for the same vertical profile at M6 is presented in Figure 15. Strong similarity is observed between the two types of measurements, down to about $z = 3$ m, at which point some divergence is observed, with the Cobra Probes being unable to capture the spike in TKE to the same extent as the PIV measurements, which may be partially due to the reverse flow at this location. Although Reynolds number effects may contribute to the higher TKE for the U14 PIV profile at $z < 3$
m, Reynolds number independence appears to be preserved completely among the three Cobra Probe profiles all the way down. This observation once again raises the question of why the measurements of the 1:25 scale model at WindEEE are higher than those over the 1:100 BLWTL model, and why they are closer to the full-scale measurements, as seen in Figure 12b at M6. The resolution of the model, and the ability to measure closer to the surface level thus appears to be one contributing factor. A separate study by Lange et al. recently submitted for publication investigates the effect of sharpening
the escarpment edge. Preliminary analysis shows that this has a large effect on the flow behaviour in the region close to the top of the hill.

### 6 Conclusions

An experimental investigation to characterize the mean and turbulent flow behaviour over a steep escarpment, represented by the topography of Bolund Hill, was conducted at two distinct scales (1:100 and 1:25) by means of wind tunnel testing using
Particle Image Velocimetry (PIV) and Cobra Probes. A range of Reynolds numbers, boundary layer inflow profiles, and





upstream roughness values were examined. At the WindEEE research facility, three uniform fan profiles and one modified shear profile were tested at two different upstream roughness configurations, for a total of eight unique sets of inflow conditions. These results, presented in the form of normalized speed-up and TKE increment, were compared to each other and to measurements from the field campaign and previous experimental work, to attempt to establish the relative contributions of the key upstream parameters to flow behaviour over the hill.

Mean flow behaviour was found to be generally resilient to changes in upstream conditions, with negligible Reynolds number dependence observed between the uniform fan speed cases, across a Reynolds number range of $1.7 \times 10^5$ to $4.6 \times 10^5$, for both Cobra Probe and PIV measurements. Slight modification of the speed-up behaviour was observed for the shear profile case, but this did not appear to be related to the Reynolds number. Lower upstream roughness was observed to cause a marginal increase in peak speed-up at the escarpment for the uniform fan speed cases, whereas for the shear case, lower roughness caused a slight reduction in speed-up, particularly near the surface. Slightly higher values of speed-up were observed for the 1:25 scale model compared to the 1:100 model, which are attributed to factors such as proximity of the instrument to the model surface, or model surface roughness.

From the turbulent flow field data represented in the form of TKE increment, a weak Reynolds number dependence was observed whereby TKE increased with an increase in the Reynolds number, but only in the highly turbulent shear layer near the escarpment. Lower upstream roughness also served to moderately increase peak TKE among the uniform fan speed cases. A much more significant TKE increase was observed for the shear profile case, where peak normalized TKE at a height of 2 m above the hill increased by over 200% compared to the uniform fan speed case at a similar Reynolds number. Through modification of the inflow shear profile, the WindEEE facility was able to produce TKE increments that were closer to full-scale measurements, and higher than those that had been achieved previously in conventional wind tunnels, indicating a promising trend for future work in characterizing flow over topography.

For the wind developer, these results reinforce the need for very careful and detailed assessment of wind turbine inflow conditions in complex topography, as even very small changes to the inflow profile used in the modelling process can cause highly significant changes at turbine height, particularly in the turbulent flow behaviour.

*Acknowledgements*: The present work is supported by NSERC, CFI and UWO, in addition to the Center for Computational Wind Turbine Aerodynamics and Atmospheric Turbulence funded by the Danish Council for Strategic Research, grant number 09-067216.



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

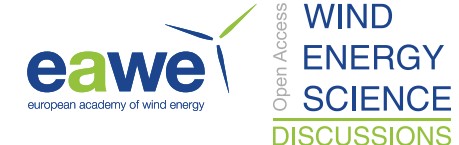

**Table 1.** Inflow parameters for WindEEE PIV test cases

| Case ID | Fan configuration | $u_h$ (m/s) | Re | $z_0$ (m) | $u_{*05}$ (m/s) Method 1 | $u_{*05}$ (m/s) Method 2 | $u_*$ (m/s) Method 3 | $u_*$ (m/s) Method 4 |
|---|---|---|---|---|---|---|---|---|
| U5RC1 | All fans 20% | 5.42 | $1.70 \times 10^5$ | $1.84 \times 10^{-3}$ | 0.314 | 0.326 | 0.254 | 0.333 |
| U5RC2 | All fans 20% | 5.49 | $1.72 \times 10^5$ | $1.96 \times 10^{-6}$ | 0.229 | 0.252 | 0.145 | 0.231 |
| U8RC1 | All fans 30% | 8.70 | $2.72 \times 10^5$ | $1.98 \times 10^{-3}$ | 0.488 | 0.489 | 0.409 | 0.505 |
| U8RC2 | All fans 30% | 8.57 | $2.68 \times 10^5$ | $4.12 \times 10^{-7}$ | 0.373 | 0.415 | 0.203 | 0.355 |
| U14RC1 | All fans 50% | 14.60 | $4.57 \times 10^5$ | $2.72 \times 10^{-3}$ | 0.856 | 0.869 | 0.723 | 0.848 |
| U14RC2 | All fans 50% | 14.69 | $4.60 \times 10^5$ | $2.29 \times 10^{-6}$ | 0.668 | 0.361 | 0.392 | 0.640 |
| U15RC1 | Fan rows 1,2,4: 50% Fan row 3: 75% | 15.60 | $5.21 \times 10^5$ | $2.87 \times 10^{-4}$ | 0.992 | 1.070 | 0.650 | 0.970 |
| U15RC2 | Fan rows 1,2,4: 50% Fan row 3: 75% | ~15.60 | ~$5.21 \times 10^5$ | Not measured* | | | | |

Roughness lengths shown in full-scale.
*The upstream profile for case U15RC2 was not measured due to an oversight, and as a result the U15RC2 PIV results had to be normalized against the U15RC1 upstream Cobra Probe data.

**Table 2.** Inflow parameters for BLWTL experiment

| Case ID | $u_h$ (m/s) | Re | $z_0$ (m) | $u_{*05}$ (m/s) Method 1 | $u_{*05}$ (m/s) Method 2 | $u_*$ (m/s) Method 3 | $u_*$ (m/s) Method 4 |
|---|---|---|---|---|---|---|---|
| BLWTL | 4.65 | $3.63 \times 10^4$ | $1.266 \times 10^{-4}$ | 0.1643 | 0.1651 | 0.1640 | 0.1858 |




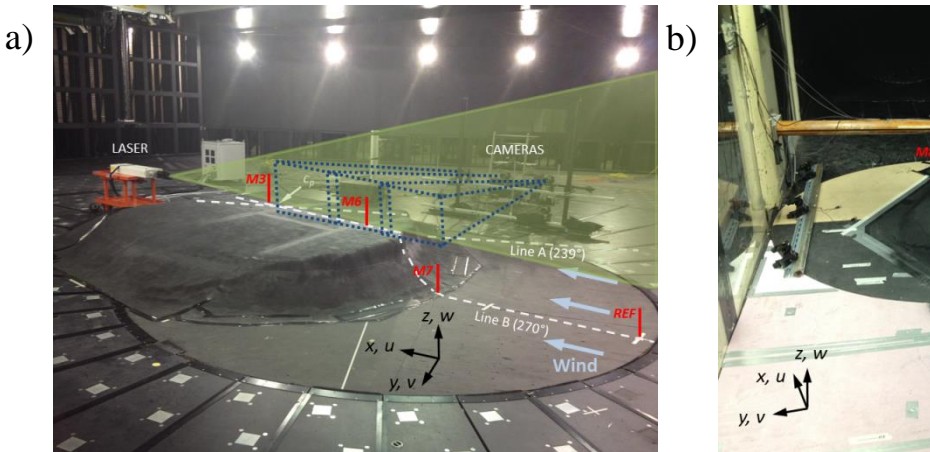
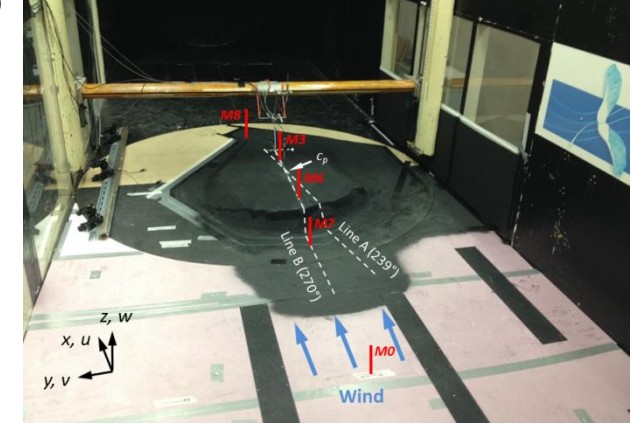

**Figure 1.** Experimental setup for a) WindEEE b) BLWTL.

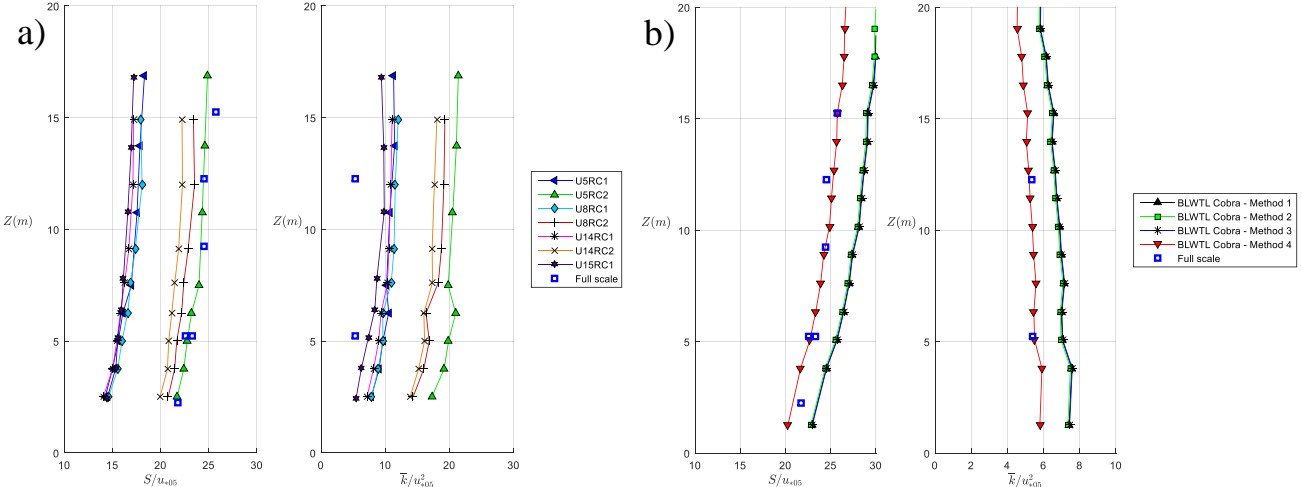

**5** **Figure 2.** a) WindEEE upstream profiles of upstream reference mean flow speed and TKE normalized by friction velocity obtained using Method 1, and b) BLWTL upstream profiles of upstream reference mean flow speed and TKE, normalized by friction velocity obtained using four different methods. S and $\overline{k}$ calculated using all three components of wind speed from Cobra Probe measurements. Z co-ordinates shown in full-scale.





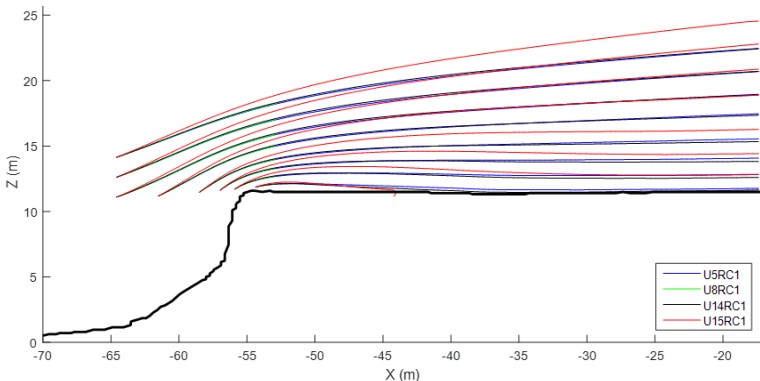

**Figure 3.** Mean streamline plots for test cases with roughness configuration RC1.

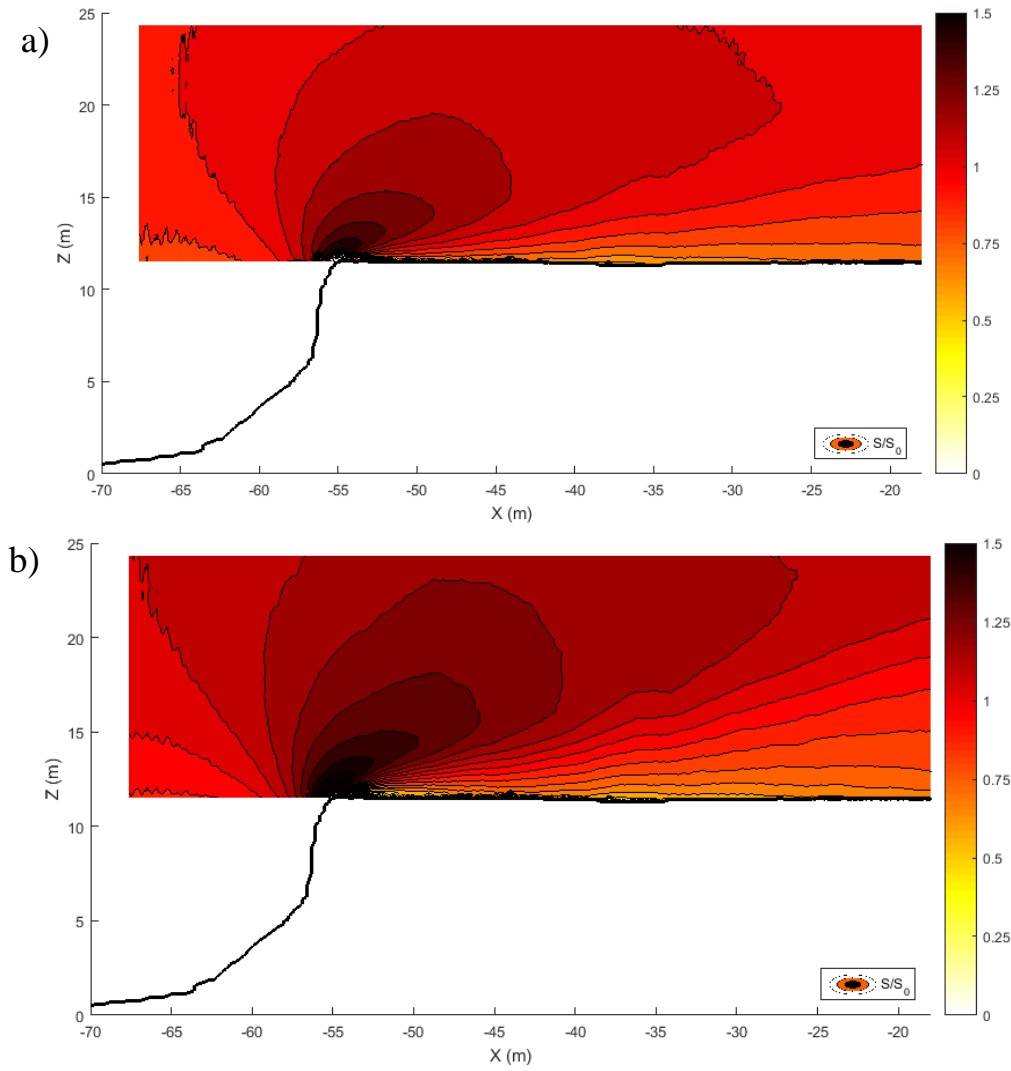

5   **Figure 4.** Speed-up ratio contour plot for a) U14RC1 and b) U15RC1.




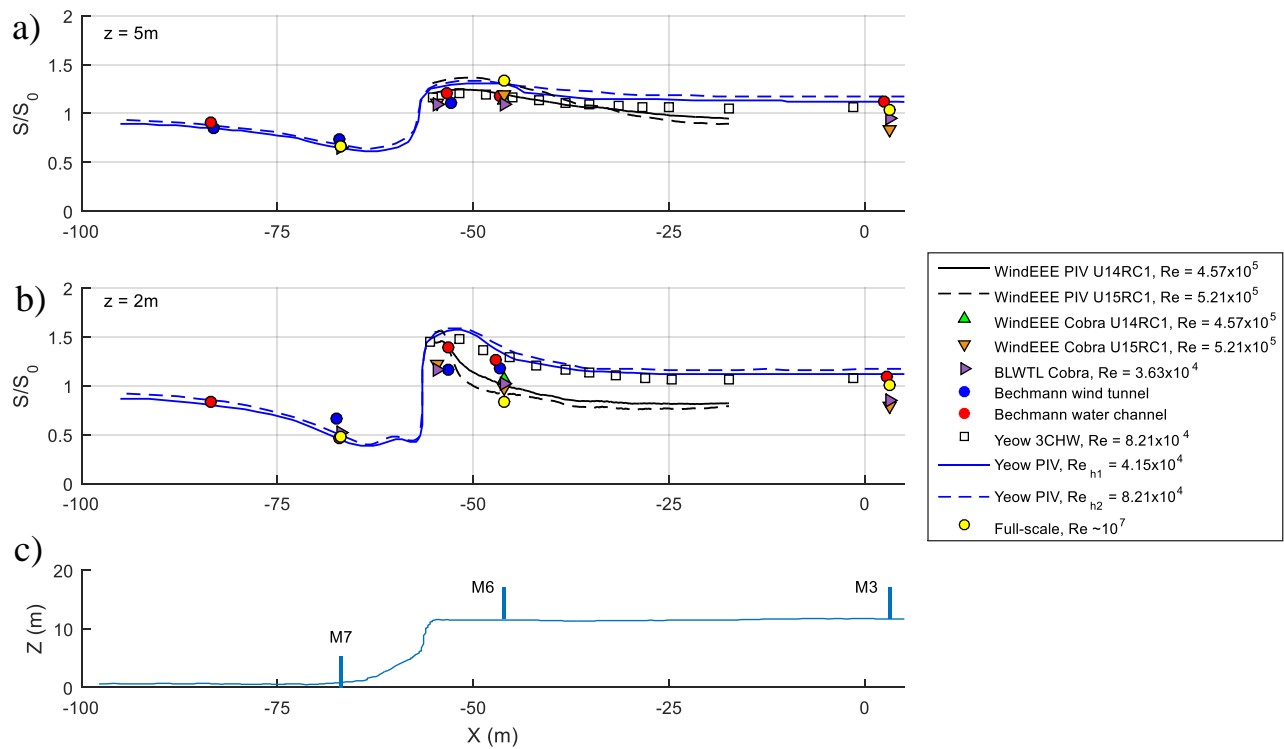

**Figure 5.** WindEEE and BLWTL speed-up ratio along horizontal profiles at a) $z = 5$ m and b) $z = 2$ m above surface level for PIV and Cobra Probe measurements with comparison to full-scale and to previous physical experiments.

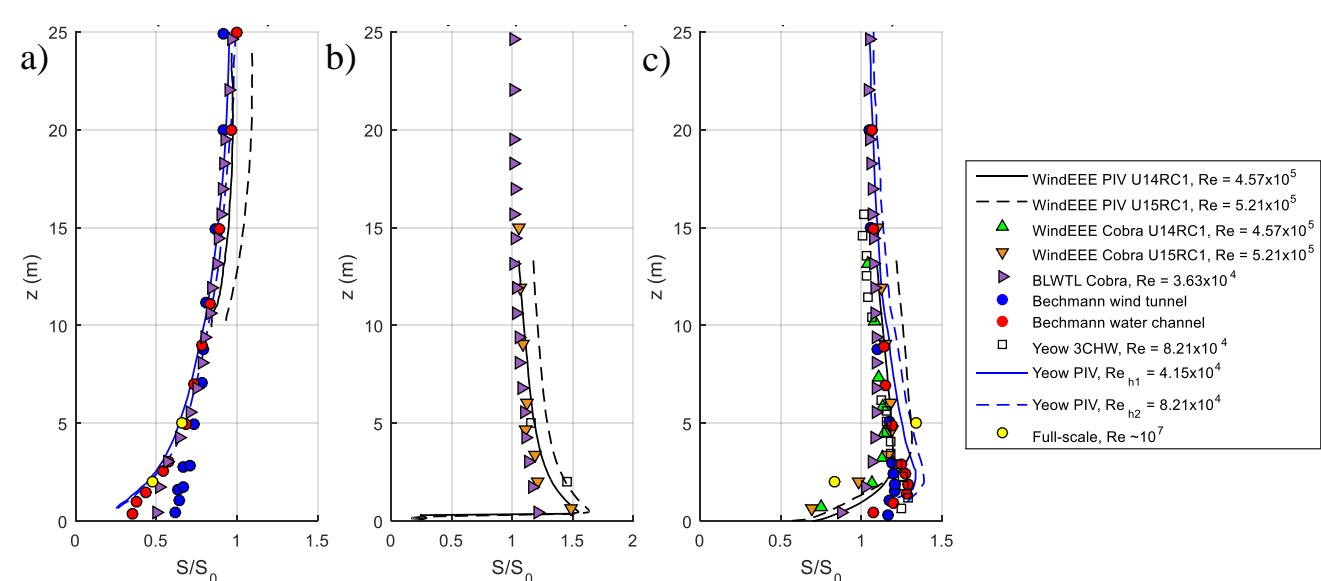

**Figure 6.** Comparison of WindEEE and BLWTL speed-up ratio along vertical profiles at a) M7 ($X = -66.9$ m), b) escarpment ($X = -54.7$ m) and c) M6 ($X = -46.1$ m) with comparison to full-scale and to previous physical experiments.




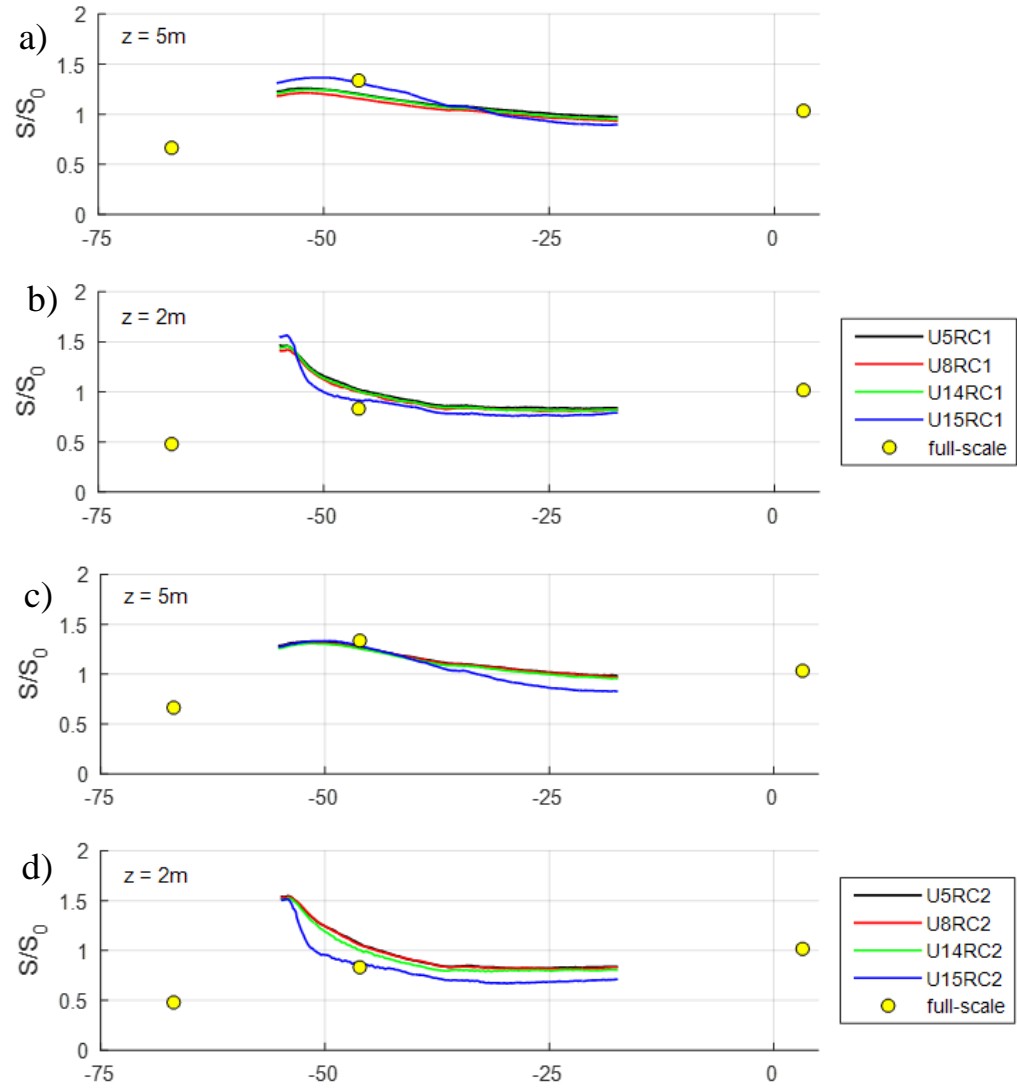

**Figure 7.** Horizontal profiles of speed-up ratio for a) RC1 cases, $z = 5$ m, b) RC1 cases, $z = 2$ m, c) RC2 cases, $z = 5$ m and d) RC2 cases, $z = 2$ m.





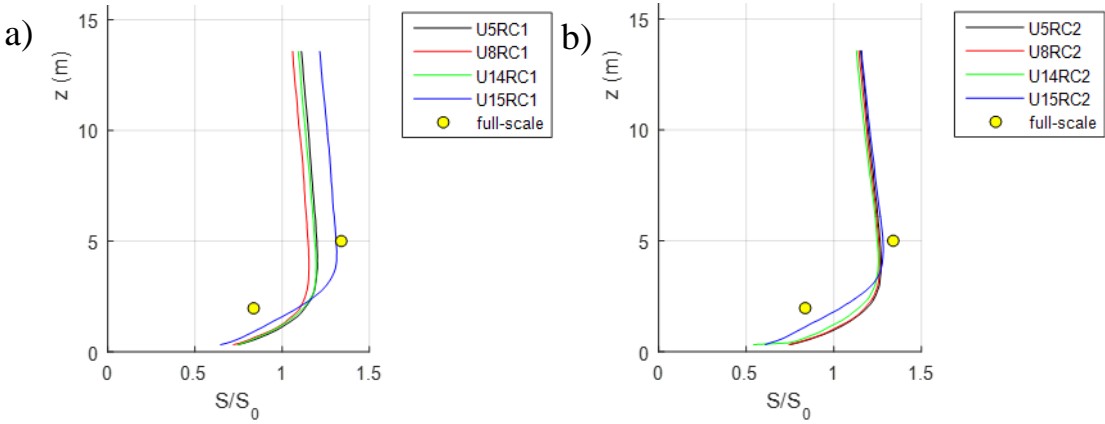

**Figure 8.** Vertical profiles of speed-up ratio at M6 for a) RC1 cases (left) and RC2 cases (right).

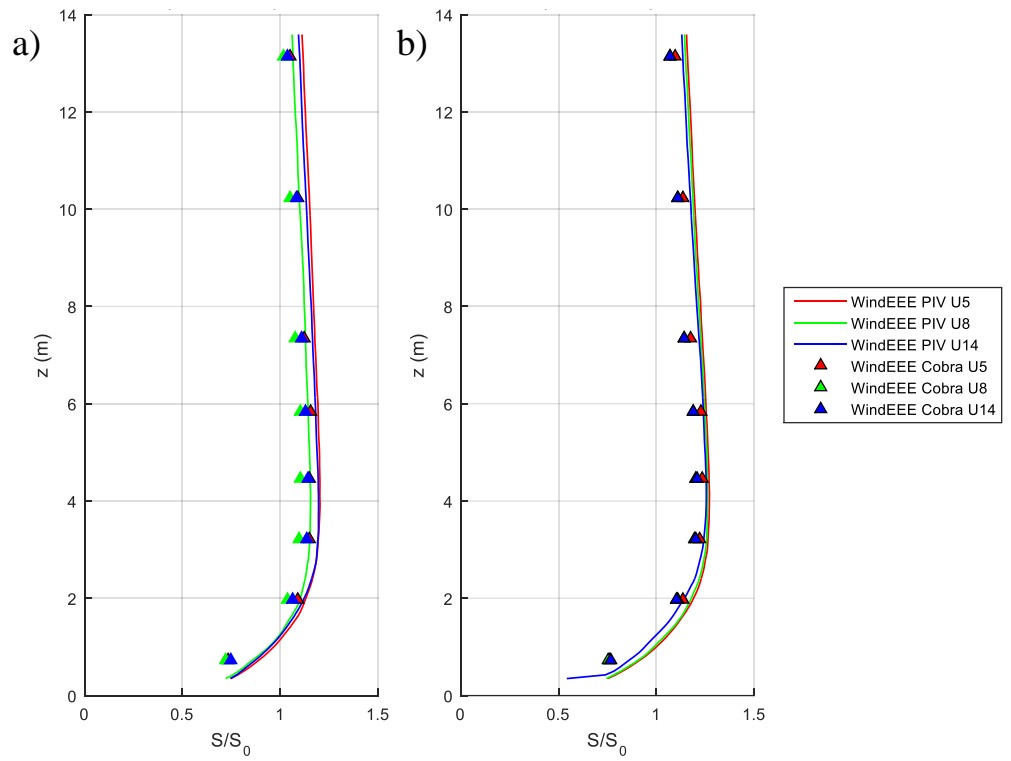

**Figure 9.** WindEEE PIV and Cobra Probe vertical profiles of speed-up ratio at M6, for RC1 (left) and RC2 (right).





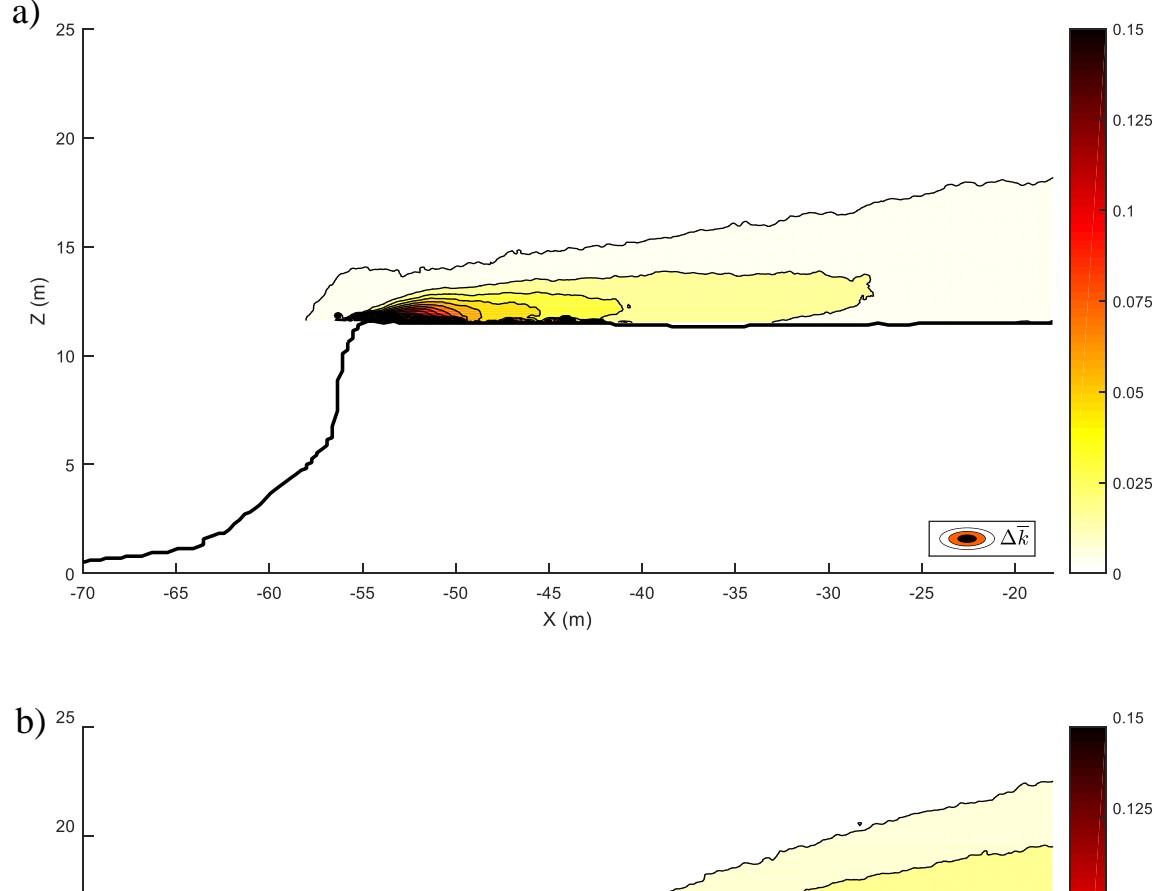

**Figure 10.** Mean TKE increment contour plot for a) U14RC1, and b) U15RC1.



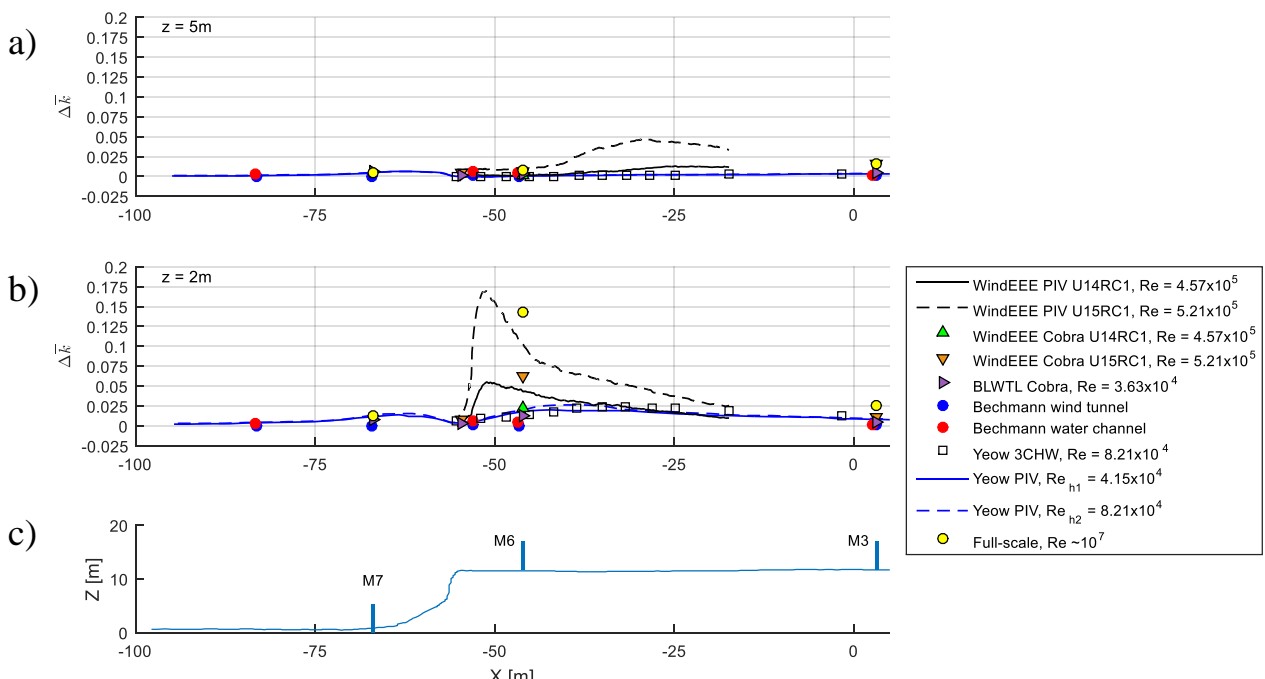

**Figure 11.** WindEEE and BLWTL TKE increment along horizontal profiles at a) $z = 5m$ and b) $z = 2m$ above surface level for PIV and Cobra Probe measurements with comparison to full-scale and to previous physical experiments.

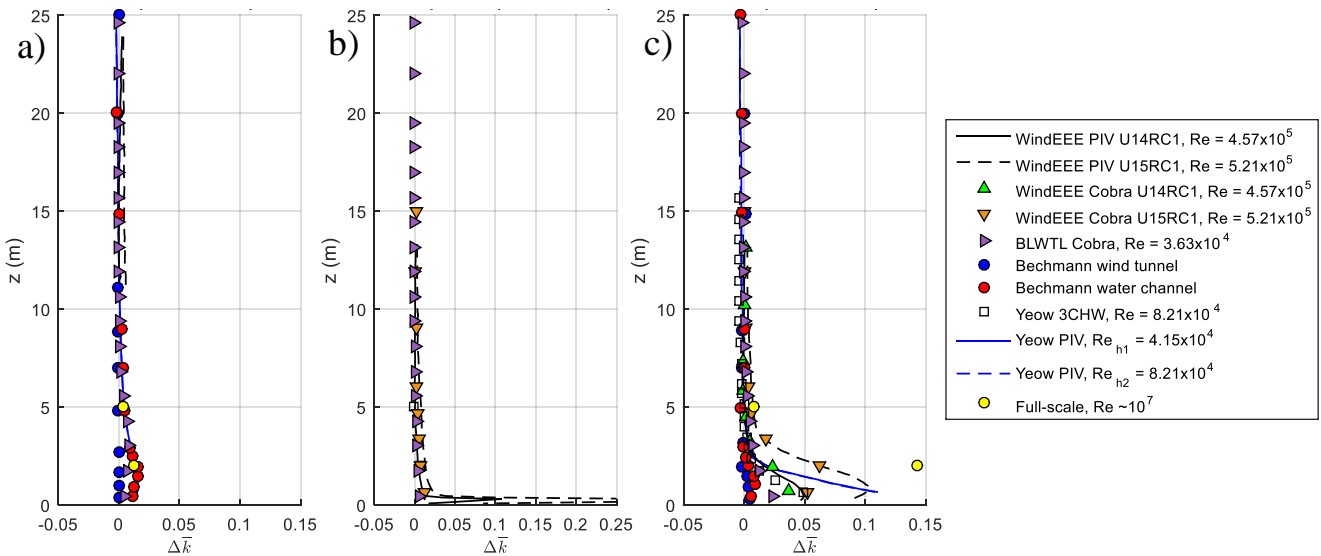

**Figure 12.** Comparison of WindEEE and BLWTL TKE increment along vertical profiles at a) M7 ($X = -66.9$ m), b) escarpment ($X = -54.7$ m) and c) M6 ($X = -46.1$ m) with comparison to full-scale and to previous physical experiments.





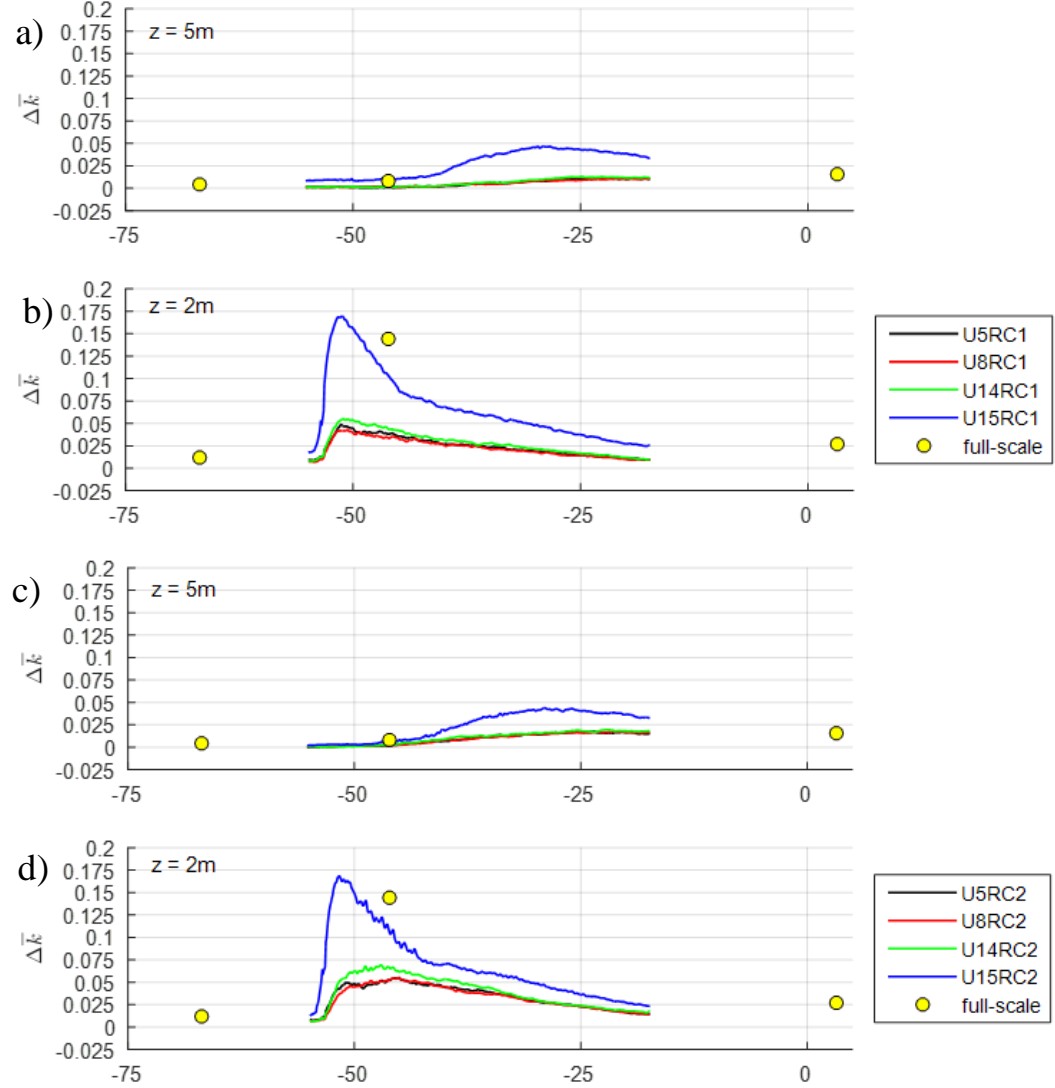

**Figure 13.** Change in TKE horizontal profiles for a) RC1 cases at $z = 5$ m. and b) RC1 cases at $z = 2$ m, c) RC2 cases at $z = 5$ m, and d) RC2 cases at $z = 2$m.



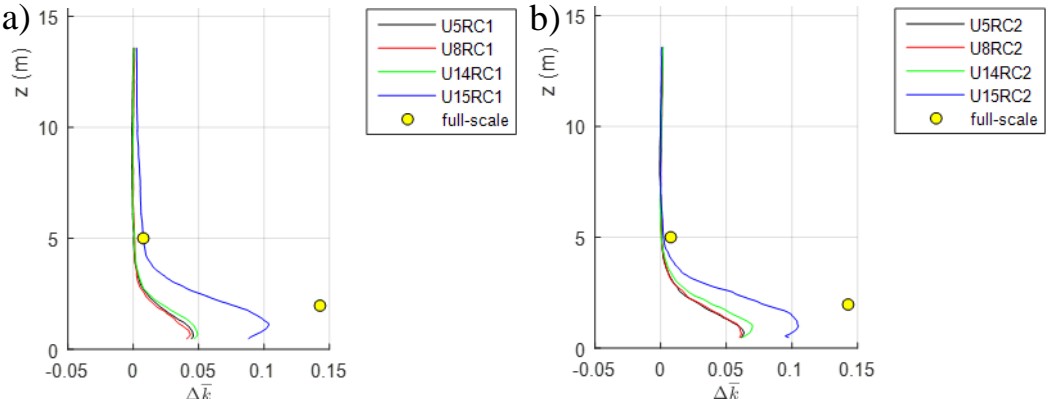

**Figure 14.** Vertical profiles of TKE increment at M6 for a) RC1 cases and b) RC2 cases.

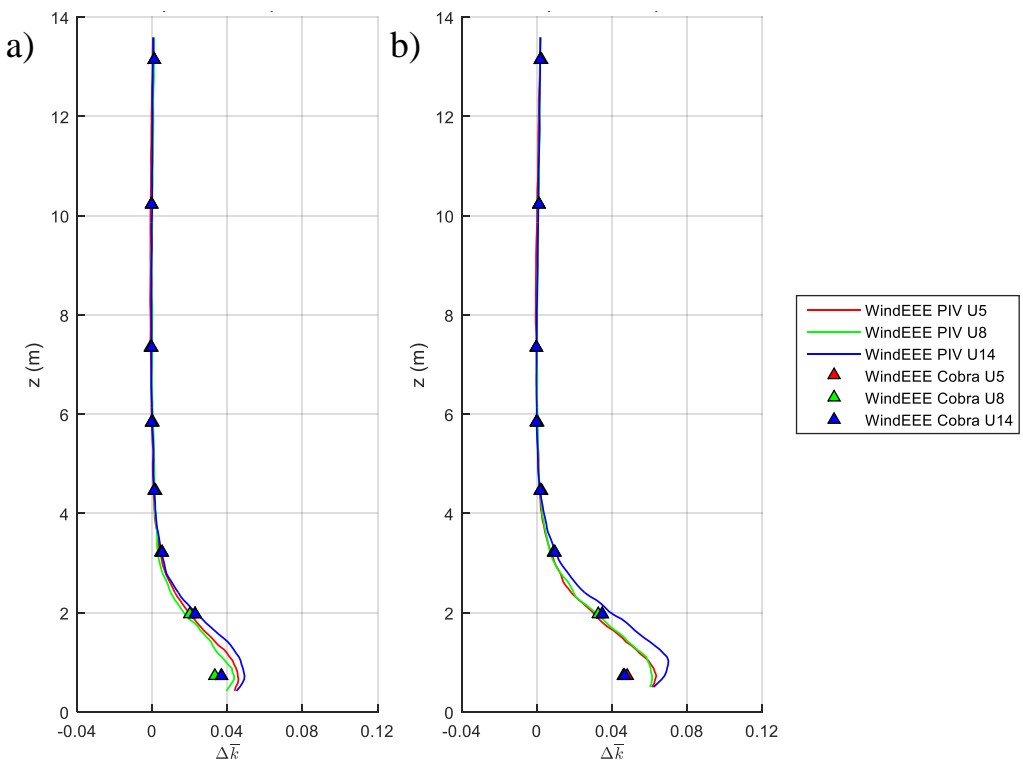

**Figure 15.** WindEEE PIV and Cobra Probe vertical profiles of TKE increment at M6, for a) RC1 and b) RC2 cases.

