# Peer review of "Effect of Reynolds number and inflow parameters on mean and turbulent flow over complex topography"

_Wind Energy Science, 2016_

## Referee Comment (RC1) · A. Cuerva (Referee) · 22 Jun 2016

**Abstract**

Substitute " ...was an escarpment" by "...was the escarpment facing westerly winds"

**2. Experimental setup**

**2.1. WindEEE experimental setup**

**2.1.1. WindEEE facility**

Too many indexing levels without text. You should be able to write something just after section 2. and subsection 2.1.

Please check the whole text.

**Page 8, line 30**

"...used by Berg et al. (2011) to calculate friction velocity using data from the upstream reference mast M0 in the Bolund field campaign". Please indicate at which height.

**Page 9, lines 15-20**

About the discussion on aerodynamic roughness length, $z_0$, for RC1 and RC2 cases, you should indicate that RC1 tests are well within the fully rough regime since $u_* z_0 \nu^{-1} \gg 1$ (so no dependence of $z_0$ on $u_*$ is expected), whereas RC2 tests are in the smooth regime since $u_* z_0 \nu^{-1} < 0.2$, considering Bowen (2003), so dependence of $z_0$ on $u_*$ is expected. As well as in the case of BLWTL test which is in the transitionally rough regime. Providing some results for the friction Reynolds number in this section is convenient.

**Page 10, lines 15-20**

Similarities with inflow profiles in Yeow et. al (2015) are expected since $u_*$ values are of the same order and $z_0$ values are almost the same in both studies (Yeow et al. 2015 and the present one).

**Page 10, lines 25-30**

Providing statistics (i.e. number of occurrences of instantaneous values $u(t) < 0$) for a given set of PIV images pairs) as in Yeow et al, 2015 would provide some insight on the statistical significance of instantaneous inverse flow ocurrence.

**Page 14, lines 25-30**

It is evident that U15 cases (both RC1 and RC2) present the best fit to full-scale data both in terms on $S/S_0$ and $\Delta \bar{k}$ at M6 (mainly at 2m a.g.l). This is one of the issues pointed out in previous works. The difference in $S/S_0$ and $\Delta \bar{k}$ PIV patterns between U14RC1 and U15RC1 are evident. The difference in the setup of the wind tunnel fans between U14 and U15 cases is also clear, but there has not been identified any significative difference in the studied non-dimensional inflow parameters.

The differences in $S/S_0$ and $\Delta \bar{k}$ at M6 (mainly at 2m a.g.l) are unlikely justified by the small change in Reynolds number, $Re_h$ between U14 and U15 cases. The difference in the ratio $h/z_0$ is neither the cause of the significantly improved match for cases U15 (both RC1 and RC2).

The question is, which are the relevant non-dimensional inflow characteristics that are affected by the difference in the wind tunnel fans setup between U14 and U15 cases, and that affect $S/S_0$ and $\Delta\bar{k}$ at M6 (mainly at 2m a.g.l)?. Have you checked the vertical inflow profiles for $\overline{uw}(z)\,u_{*05}^{-1}$ and integral length scales $L_{u_i}^x(z)\,h^{-1}$ ?. Are there any difference between U15 cases and the other ones for these paramaters?.

**Page 19, table 1**

Full scale data should be included.

**References**

Bowen, A. J., 2003. Modelling of Strong Wind Flows over Complex Terrain at Small Geometric Scales. Journal of Wind Engineering and Industrial Aerodynamics 91, 1859–1871.

---

## Referee Comment (RC2) · Anonymous Referee #2 · 11 Jul 2016

General comments:

Laboratory experiments at two scales were performed to investigate Reynolds number effects, and other parameters, on the mean flow and turbulence levels around a complex topography. The experimental setup is good, and measurements with PIV and Cobra probes are well complemented. Some clarifications are needed to improve the impact and quality of the paper.

Specific comments:

- Just suggestion: revise the title, maybe something more compact.

- Consider including a general schematic of the setup with the measurement locations

of the PIV and probes.

- Discuss about the boundary layer and topography ratio in the two experimental setups. If possible link that with the field.

- Use the PIV measurements to show more features of the flow field. Use this information to get more insight on the effect of the complex topography.

- The streamlines of figure 3 are not very informative. You might consider adding more details. It is just a suggestion.

- Line 15: clarify the meaning of line B at that point.

- Line 29: specify the meaning of 'boundary-layer flows at larger laboratory scales'. . .i.e., larger respect to ...?

- Please provide more details on the PIV, including vector spacing, % of overlapping, particle seeding, and laser thickness.

- The use of equation 2 to extrapolate probes/PIV is not clear. The log law is valid in a limited region of the flow (constant stress zone). Is this the case?

- Revise equation 4.

- The estimation of the u* from Method 1 and 2 using a reference height is ok if such height is within the constant stress zone. Please verify this.

- Consider using specific figures to show the estimation of u* from the log law.

- Line 30: the discussion of the small region with some negative velocity vectors is not supported from figure 3.

- In general, you might consider discussing relative differences to infer if variables are similar or not.

- Page 13, line 20: you use a reference to support Re independence to explain the differences of the profiles, but the spirit of the paper is to evaluate Re effects! Please

clarify this apparent confusion.

Technical corrections:

- Please proof-read the text, and remove inconsistencies like the setting in figure 5 and text in page 11, line 22. Also revise figure captions.

- Replace upstream by upwind,

---

## Author Response (AR1)

**wes-2016-17: Response to reviewer comments**

**Reply to the comments of reviewer RC1**

5   We are thankful to the reviewer for reviewing the manuscript. Below is a point-by-point reply to the comments.

**Abstract**
Substitute " ...was an escarpment" by "...was the escarpment facing westerly winds"

10   *Action: Text will be modified as per the comment.*

**2. Experimental setup**
**2.1. WindEEE experimental setup**
**2.1.1. WindEEE facility**
Too many indexing levels without text. You should be able to write something just after section 2. and
15   subsection 2.1.
Please check the whole text.

*Action: The third heading level will be removed, and additional text will be inserted between the first and second heading*

*levels for clarity. The rest of the text will be check and adjusted as necessary.*

**Page 8, line 30**
20   "...used by Berg et al. (2011) to calculate friction velocity using data from the upstream reference mast M0
in the Bolund field campaign". Please indicate at which height.

*Reply: A height of $Z = 5$ m above sea level was used by Berg et al. for this calculation.*

*Action: This will be indicated in the text.*

**Page 9, lines 15-20**
25   About the discussion on aerodynamic roughness length, $z_0$, for RC1 and RC2 cases, you should indicate
that RC1 tests are well within the fully rough regime since $u_* z_0 v^{-1}$ (so no dependence of $z_0$ on $u_*$ is
expected), whereas RC2 tests are in the smooth regime since $u_* z_0 v^{-1} < 0.2$, considering Bowen (2003), so dependence of $z_0$ on
$u_*$ is expected. As well as in the case of BLWTL test which is in the transitionally rough regime. Providing some results for
the friction Reynolds number in this section is convenient.

*Action: We will revise the text as per reviewer's suggestion regarding the rough and smooth regimes. The friction Reynolds*

*number will be indicated in the text for various cases along with the corresponding roughness regime.*

**Page 10, lines 15-20**
35   Similarities with inflow profiles in Yeow et. al (2015) are expected since $u_*$ values are of the same order and $z_0$ values are
almost the same in both studies (Yeow et al. 2015 and the present one).

*Action: The text will be revised to discuss similarities with the results of Yeow et al. (2015) based on u\* and z0 values*

**Page 10, lines 25-30**

Providing statistics (i.e. number of occurrences of instantaneous values $u(t) < 0$ for a given set of PIV images pairs) as in Yeow et al, 2015 would provide some insight on the statistical significance of instantaneous inverse flow occurrence.

*Reply:* The contours of the probability of occurrences of negative instantaneous velocity vectors in the stream-wise direction are plotted in Figures 1, 2 and 3 for three cases with different upstream conditions. Note that the colorbar is on the log scale hence, -1 corresponds to 10-1 and so on. As shown in the figures, the U15RC1 shear case (Figure 3) has relatively higher occurrences of negative vectors over a wider region above the hill surface compared to U14RC1 (Figure 1) and U14RC2 (Figure 2) cases, which were similar to each other. It is also evident in all figures that except for a very small near-surface region close to the escarpment leading edge, the probability of negative occurrences is less than 1%, which further decreases with an increase in the downstream distance and height. This is consistent with the results presented by Yeow et al. (2015).

As mentioned above, there is a very small near-surface region in the vicinity of the escarpment edge with higher probability of negative occurrences. To illustrate this, the contours of negative occurrence probability greater than 0.1 are plotted in Figure 4. As shown, the probability of occurrences of negative vectors reaches up to 80%. Considering relatively high uncertainty in the velocity measurements in the immediate vicinity of the surface, we may not be able to draw any firm conclusion about the presence of the recirculation region.

*Action:* It is of the authors' view that providing additional figures in the present paper on this topic would not be beneficial, however the text will be revised to discuss the probability of occurrences of negative vectors and its comparison with that of Yeow et al. (2015).

[Figure]

**Figure 1.** Contours of probability of occurrence of negative stream-wise instantaneous velocity (log scale), U14RC1.

[Figure]

**Figure 2.** Contours of probability of occurrence of negative stream-wise instantaneous velocity (log scale), U14RC2.

[Figure]

**Figure 3.** Contours of probability of occurrence of negative stream-wise instantaneous velocity (log scale), U15RC1.

[Figure]

**Figure 4.** Contours of probability of occurrence of negative stream-wise instantaneous velocity (linear scale), U14RC1.

**Page 14, lines 25-30**

5     It is evident that U15 cases (both RC1 and RC2) present the best fit to full-scale data both in terms on $S/S_0$ and $\Delta k$ at M6 (mainly at 2m a.g.l). This is one of the issues pointed out in previous works. The difference in $S/S_0$ and $\Delta k$ PIV patterns between U14RC1 and U15RC1 are evident. The difference in the setup of the wind tunnel fans between U14 and U15 cases is also clear, but there has not been identified any significative difference in the studied non-dimensional inflow parameters.

    The differences in $S/S_0$ and $\_k$ at M6 (mainly at 2m a.g.l) are unlikely justified by the small change in Reynolds number, 10  $Re_h$ between U14 and U15 cases. The difference in the ratio $h/z_0$ is neither the cause of the significantly improved match for cases U15 (both RC1 and RC2).

    The question is, which are the relevant non-dimensional inflow characteristics that are affected by the difference in the wind tunnel fans setup between U14 and U15 cases, and that affect $S/S_0$ and $\Delta k$ at M6 (mainly at 2m a.g.l)?. Have you checked the vertical inflow profiles for $uw(z)\, u_{-1\_05}$ and integral length scales $Lx_{ui}(z)\, h_{-1}$ ?. Are there any difference between U15 cases 15  and the other ones for these parameters?

*Reply: The reviewer has raised a very important and valid point. We have further investigated this issue and computed other flow-related parameters that can provide some insight into causes of these differences. One parameter that has a direct influence on the turbulence behavior and momentum transport is the Reynolds stress. The vertical profiles of the inflow* 20  *Reynolds shear stress are shown in Figure 5 for all cases versus the height presented in the full-scale co-ordinates. For the uniform fan configuration, as expected, the magnitude of Reynolds shear stress increased with increasing wind speed. Further, the Reynolds stress magnitude was larger in the presence of higher roughness, also an expected trend.*

*All cases show a similar trend i.e. the Reynolds stress magnitude increased towards the surface and then became almost constant, which corresponds to the constant shear stress region. As the figure shows, the constant stress region covered a* 25  *height of almost 5 m (in full scale coordinates) from the surface.*

*The results for the U15RC1 case, where the inflow shear profiles were modified, showed a different behavior locally. A local region of high shear stress is found in the height range of 10-13 m which is likely attributable to the higher operating speed of Fan row 3. This height also corresponds to the region immediately above the hill height. The Reynolds shear stress profiles normalized by the friction velocity (Method 1) are plotted in Figure 6.*

30  *The results show relatively similar normalized magnitude for all cases except the higher Reynolds stress for U15 in the high shear region. The results show that the magnitude of the non-dimensional Reynolds shear stress for U15 is about 60% higher than that for the U14 case in this high shear region.*

*The results in Figures 5 and 6 provide clear evidence of the high shear injected into the inflow for the U15 case at the hill height.*

[Figure]

***Figure 5.*** *Magnitudes of inflow Reynolds shear stress, all cases*

[Figure]

***Figure 6.*** *Inflow Reynolds shear stress normalized by u\* (Method 1), RC1 cases*

5   *Another parameter that provides characteristics of the inflow is the integral length scale. The inflow integral time scale was first computed using the Cobra probe data due to higher time resolution. The integral length scale was then computed using Taylor's hypothesis. The inflow integral length scale at different heights for all cases are plotted in Figures 7 and 8 in dimensional and dimensionless forms, respectively. There is a noticeable trend of increased length scale as Reynolds number is increased among the uniform profiles (see Figure 7), however the U15 case (shear profile) with the highest Reynolds number*

*has the lowest length scales of all RC1 cases. The difference is particularly large at hill height (Z = 11.73 m). This shows that*

*for the modified shear profile case (U15), the inflow is comprised of smaller sized eddies compared to the uniform flow cases.*

*Based on the results presented in Figures 5-8, we can infer that the inflow profile for the shear case is characterized by eddies*

*with smaller average size and higher levels of momentum transport than for the uniform fan cases particularly at the hill*

5   *height.*

*Action: These results will be discussed in the revised manuscript to demonstrate the differences between the inflow conditions*

*of the shear and uniform flow cases.*

[Figure]

**Figure 7.** Magnitudes of the inflow integral length scale, all cases

[Figure]

Figure 8. Inflow integral length scale normalized by hill height, RC1 cases

**Page 19, table 1**
Full scale data should be included.

*Action: Full scale data will be added to the table.*

**Reply to the comments of reviewer RC2**

We are thankful to the reviewer for reviewing the manuscript. Below is a point-by-point reply to the comments.

- Just suggestion: revise the title, maybe something more compact.

*Action: The title will be revised to "Effect of Reynolds number and inflow parameters on mean and turbulent flow over complex topography"*

- Consider including a general schematic of the setup with the measurement locations of the PIV and probes.

*Action: A schematic diagram will be added showing the relevant positions.*

- Discuss about the boundary layer and topography ratio in the two experimental setups.
If possible link that with the field.

*Reply: Boundary layer height is generally $3.3 < \delta/h < 3.6$ for all WindEEE cases. For BLWTL, $\delta/h$ is higher, roughly 10.5. Full scale ABL height is not indicated explicitly. If it is ~1 km, then $\delta/h \sim 85$.*

*Action: The boundary layer heights for the two setups will be included in the revised manuscript.*

- Use the PIV measurements to show more features of the flow field. Use this information
to get more insight on the effect of the complex topography.

*Reply: The content of this paper was intentionally limited to discussion on the mean and first order turbulent statistics. A second paper is currently in preparation that investigates in more detail the structure of the turbulent flow. For example, the use of Proper Orthogonal Decomposition (POD) to identify coherent structures in the flow, as well as analysis of higher order turbulent statistics. Hence, such details are beyond the scope of this paper.*

- The streamlines of figure 3 are not very informative. You might consider adding more
details. It is just a suggestion.

*Reply: The streamline plots are merely meant to provide a qualitative illustration of the mean flow behaviour: they indicate regions of flow acceleration and deceleration, a lack of recirculation, the similarity among the uniform cases, and the minor differences between the uniform and shear case.*

- Line 15: clarify the meaning of line B at that point.

*Reply: The Line B corresponds to the wind flow direction of 270o. This is already indicated in the manuscript in Section 1.2.*

- Line 29: specify the meaning of 'boundary-layer flows at larger laboratory
scales': : :i.e., larger respect to ...?

*Reply: It was in comparison with conventional wind tunnels.*

*Action: Text will be revised to include this clarification*

- Please provide more details on the PIV, including vector spacing, % of overlapping,
particle seeding, and laser thickness.

*Reply: Vector spacing was 16 pixels. At spatial resolution of 0.1902 mm/px, this corresponds to roughly 3.04 mm. 75% overlap was used for the interrogation window cross-correlation.*

*Seeding particle size is estimated to be ~5 μm based on the estimate by Ayotte and Hughes (2004) for a commercial fog generator similar to the one used in the present study. These values will be included in the text.*

10 *The laser beam diameter specified by the supplier is 9.5 mm. Use of laser optics makes the resulting laser sheet thickness more difficult to estimate however it is expected to be similar.*

*Action: The above information will be included in the text.*

- The use of equation 2 to extrapolate probes/PIV is not clear. The log law is valid in a
15 limited region of the flow (constant stress zone). Is this the case?

*Reply: Figure 12 shows the inflow wind speed for the measured Cobra Probe data, the log law profile, and the range of heights corresponding to the PIV measurement window. The U14RC1 case is shown; the others are similar.*

*From the plots of inflow Reynolds stress, the region of constant shear stress extends only to $Z \approx 5$ m. Ideally the inflow profile*
20 *would have been measured up to the maximum height of the PIV measurement window, however this was not feasible with the experimental setup available at the time. Thus, the logarithmic law was selected as the best fit to the measured data, given the limitations described.*

*Action: Text will be modified to further clarify the procedure taken*

[Figure]

**Figure 1**. *Extrapolation of measured Cobra probe inflow wind speed, U14RC1*

- Revise equation 4.

*Action: The equation will be revised to include the missing overbar on the second "k".*

- The estimation of the u* from Method 1 and 2 using a reference height is ok if such height is within the constant stress zone. Please verify this.

*Reply: The inflow Reynolds shear stress profiles are plotted for all cases in Figure 2. As shown, the height of Z = 5 m (reference height) is within the constant stress zone.*

[Figure]

**Figure 2.** Magnitudes of inflow Reynolds shear stress, all cases

*Action: Text will be revised for clarity.*

- Consider using specific figures to show the estimation of u* from the log law

*Reply: Estimation of u\* was conducted using the standard approach of determining the linear line of best fit on a semi-log plot, where the slope of the line is k/u\* and the y-intercept is ln(z0). Figure 3 shows this plot for all of the inflow Cobra Probe measurements, while Figure 4 shows only the subset of data within the region of constant shear stress. It is the best fit line between data points in Figure 4 that was used to estimate u\* and z0 for Method 3 in the paper.*

*This figure was not included in the paper given that the procedure is fairly standard, and in order to keep the total number of figures to a reasonable number.*

*Action: Text will be added or modified to clarify the approach used.*

[Figure]

**Figure 3.** Semi-log plot of inflow stream-wise velocities, all measured data.

[Figure]

**Figure 4.** Semi-log plot of inflow stream-wise velocities, subset of data within region of constant shear stress.

- Line 30: the discussion of the small region with some negative velocity vectors is not

supported from figure 3.

*Reply: Considering relatively high uncertainty in the velocity measurements in the immediate vicinity of the surface, we may not be able to draw any firm conclusion about the presence of the recirculation region.*

*Action: The text in this section will be modified to remove any reference to the recirculation region where negative velocity exists.*

- In general, you might consider discussing relative differences to infer if variables are similar or not.

*Action: The percent difference between results for different cases will be added to the text where it has not been done already.*

- Page 13, line 20: you use a reference to support Re independence to explain the differences of the profiles, but the spirit of the paper is to evaluate Re effects! Please clarify this apparent confusion.

*Reply: The reference was not intended to explain the difference between profiles, but rather to indicated that the present results for the uniform cases agree with the literature (lack of Reynolds number dependence), and thus another variable besides Reynolds number appears to be responsible for the differences between profiles.*

*Action: Text will be revised to clarify this*

Technical corrections:
- Please proof-read the text, and remove inconsistencies like the setting in figure 5 and text in page 11, line 22. Also revise figure captions.

*Action: The entire text will be proofread for any inconsistencies and adjusted accordingly.*

- Replace upstream by upwind,

*Action: This will be replaced.*

**Reply to Editor Comments**

- Ensure that all text on figures (e.g. Figure 1) can be read
- Ensure that all acronyms are defined (e.g. CNC)
- Ensure capitalization of variables is consistent (u (pg 6 line 19) vs U (pg 6 line 24) )
- Consider increasing font size for Figures 2, 3.

*Action: The text in the final manuscript will be adjusted to account for each of the above comments*

[revised manuscript text omitted]
$ | $1.70\times10^5$ | $1.72\times10^5$ | $2.72\times10^5$ | $2.68\times10^5$ | $4.57\times10^5$ | $4.60\times10^5$ | $5.21\times10^5$ | $3.63\times10^4$ | $4.25-10.2\times1$ |
| $z_0$ (m) | $1.84\times10^{-3}$ | $1.96\times10^{-6}$ | $1.98\times10^{-3}$ | $4.12\times10^{-7}$ | $2.72\times10^{-3}$ | $2.29\times10^{-6}$ | $2.87\times10^{-4}$ | $1.27\times10^{-4}$ | $6\times10^{-4}$ |
| $u_{*05}$ (m/s) Method 1 | 0.314 | 0.229 | 0.488 | 0.373 | 0.856 | 0.668 | 0.992 | 0.164 | 0.47 |
| $u_{*05}$ (m/s) Method 2 | 0.326 | 0.252 | 0.489 | 0.415 | 0.869 | 0.361 | 1.070 | 0.165 | — |
| $u_*$ (m/s) Method 3 | 0.254 | 0.145 | 0.409 | 0.203 | 0.723 | 0.392 | 0.650 | 0.164 | — |
| $u_*$ (m/s) Method 4 | 0.333 | 0.231 | 0.505 | 0.355 | 0.848 | 0.640 | 0.970 | 0.186 | — |
| $u_* z_0 v^{-1}$ | 37.1 | 0.03 | 62.1 | 0.01 | 149.5 | 0.10 | 18.3 | 1.34 | 18 |
| $I_{u_{h0}}$ | 0.132 | 0.127 | 0.131 | 0.131 | 0.130 | 0.126 | 0.137 | 0.096 | 0.12 |
| $\delta/h$ | 3.58 | 3.37 | 3.62 | 3.37 | 3.58 | 3.28 | 3.45 | 10.6 | — |

Roughness lengths shown in full-scale. The inflow profile for case U15RC2 was not measured due to an oversight, and as a result the U15RC2 PIV results were normalized against the U15RC1 inflow data.

[Figure]

**Figure 1.** Schematic of WindEEE experimental setup.

[Figure]

[Figure]

**Figure 2.** Photograph of the WindEEE experimental setup.

[Figure]

**Figure 3.** Photograph of the BLWTL experimental setup.

[Figure]

**Figure 4.** a) WindEEE inflow profiles of a) mean flow speed and b) TKE normalized by friction velocity obtained using Method 1, S and $\bar{k}$ calculated using all three components of wind speed from Cobra Probe measurements. $Z$ co-ordinates shown in full-scale units.

[Figure]

5 **Figure 5.** WindEEE inflow profiles: a) Reynolds shear stress magnitudes b) Reynolds shear stress normalized by friction velocity (Method 1), and c) integral length scales of the streamwise velocity component, normalized by hill height.

¶
¶

[Figure]

**Figure 6.** BLWTL inflow profiles of a) mean flow speed and b) TKE, normalized by friction velocity obtained using four different methods.

[Figure]

5    **Figure 7.** Streamlines of mean velocity for various cases with roughness configuration RC1.

[Figure]

**Figure 8.** Contours of speed-up ratio $(S/S_0)$ for a) U14RC1 and b) U15RC1.

[Figure]

**Figure 9.** Horizontal profiles of speed-up ratio from PIV and Cobra Probe measurements (WindEEE and BLWTL cases) at a) $z = 5$ m and b) $z = 2$ m above hill surface. Results from full-scale and other previous physical experiments presented for comparison.

[Figure]

**Figure 10.** Vertical profiles of speed-up ratio for WindEEE and BLWTL cases at a) M7 ($X = -66.9$ m), b) escarpment ($X = -54.7$ m) and c) M6 ($X = -46.1$ m). Results from full-scale and other previous physical experiments presented for comparison.

[Figure]

**Figure 11.** Horizontal profiles of speed-up ratio for a) RC1 cases, $z = 5$ m, b) RC1 cases, $z = 2$ m, c) RC2 cases, $z = 5$ m and d) RC2 cases, $z = 2$ m.

[Figure]

**Figure 12.** Vertical profiles of speed-up ratio at M6 for a) RC1 cases and b) RC2 cases

[Figure]

**Figure 13.** Vertical profiles of speed-up ratio from WindEEE PIV and Cobra Probe at M6, for a) RC1 and b) RC2

[Figure]

**Figure 14. Contours of** mean TKE increment $\Delta \bar{k}$ for a) U14RC1, and b) U15RC1.

[Figure]

**Figure 15.** Horizontal profiles of TKE increment $\Delta \bar{k}$ from PIV and Cobra Probe measurements (WindEEE and BLWTL cases) at a) $z = 5$ m and b) $z = 2$ m above hill surface. Results from full-scale and other previous physical experiments shown for comparison.

[Figure]

**Figure 16.** Vertical profiles of TKE increment $\Delta \bar{k}$ from PIV and Cobra Probe measurements (WindEEE and BLWTL cases) at a) M7 ($X = $ -66.9 m), b) escarpment ($X = $ -54.7 m) and c) M6 ($X = $ -46.1 m). Results from full-scale and other previous physical experiments shown for comparison.

[Figure]

**Figure 17.** Horizontal profiles of TKE increment $\Delta\bar{k}$ for a) RC1 cases at $z = 5$ m, b) RC1 cases at $z = 2$ m, c) RC2 cases at $z = 5$ m, and d) RC2 cases at $z = 2$m.

[Figure]

**Figure 18.** Vertical profiles of TKE increment $\Delta\bar{k}$ at M6 for a) RC1 cases and b) RC2 cases.

[Figure]

**Figure 19.** Vertical profiles of TKE increment $\Delta\bar{k}$ from WindEEE PIV and Cobra Probe at M6, for a) RC1 and b) RC2 cases.

**Page 22: [1] Deleted**        **Ryan Kilpatrick**        **2016-09-14 11:42:00 PM**

Friction velocity is still calculated using Eq. 6, however unlike in Method 1 where a single data point was used, Method 3 uses the mean of the values within the constant shear stress region, which were identified from the plots of height vs. $\overline{u'w'}$ as the first three data points closest to the floor for each test run.

**Page 33: [2] Deleted**        **Ryan Kilpatrick**        **2016-09-12 5:26:00 PM**

| Case ID | Fan configuration | $u_h$ (m/s) | Re | $z_0$ (m) | $u_{*05}$ (m/s) Method 1 | $u_{*05}$ (m/s) Method 2 | $u_*$ (m/s) Method 3 | $u_*$ (m/s) Method 4 |
|---|---|---|---|---|---|---|---|---|
| U5RC1 | All fans 20% | 5.42 | $1.70 \times 10^5$ | $1.84 \times 10^{-3}$ | 0.314 | 0.326 | 0.254 | 0.333 |
| U5RC2 | All fans 20% | 5.49 | $1.72 \times 10^5$ | $1.96 \times 10^{-6}$ | 0.229 | 0.252 | 0.145 | 0.231 |
| U8RC1 | All fans 30% | 8.70 | $2.72 \times 10^5$ | $1.98 \times 10^{-3}$ | 0.488 | 0.489 | 0.409 | 0.505 |
| U8RC2 | All fans 30% | 8.57 | $2.68 \times 10^5$ | $4.12 \times 10^{-7}$ | 0.373 | 0.415 | 0.203 | 0.355 |
| U14RC1 | All fans 50% | 14.60 | $4.57 \times 10^5$ | $2.72 \times 10^{-3}$ | 0.856 | 0.869 | 0.723 | 0.848 |
| U14RC2 | All fans 50% | 14.69 | $4.60 \times 10^5$ | $2.29 \times 10^{-6}$ | 0.668 | 0.361 | 0.392 | 0.640 |
| U15RC1 | Fan rows 1,2,4: 50% Fan row 3: 75% | 15.60 | $5.21 \times 10^5$ | $2.87 \times 10^{-4}$ | 0.992 | 1.070 | 0.650 | 0.970 |
| U15RC2 | Fan rows 1,2,4: 50% Fan row 3: 75% | ~15.60 | ~$5.21 \times 10^5$ | Not measured* | | | |

**Page 33: [3] Deleted**        **Ryan Kilpatrick**        **2016-09-12 5:28:00 PM**

**Table 2.** Inflow parameters for BLWTL experiment

b)

| Case ID | $u_h$ (m/s) | Re | $z_0$ (m) | $u_{*05}$ (m/s) Method 1 | $u_{*05}$ (m/s) Method 2 | $u_*$ (m/s) Method 3 | $u_*$ (m/s) Method 4 |
|---------|-------------|-----|-----------|--------------------------|--------------------------|----------------------|----------------------|
| BLWTL | 4.65 | $3.63 \times 10^4$ | $1.266 \times 10^{-4}$ | 0.1643 | 0.1651 | 0.1640 | 0.1858 |